# An enhancer cluster controls gene activity and topology of the *SCN5A-SCN10A* locus in vivo

Joyce C.K. Man[1], Rajiv A. Mohan[1,3], Malou van den Boogaard[1,3], Catharina R.E. Hilvering[2], Catherine Jenkins[1], Vincent Wakker[1], Valerio Bianchi [2], Wouter de Laat[2], Phil Barnett[1], Bastiaan J. Boukens[1] & Vincent M. Christoffels[1]*

Mutations and variations in and around *SCN5A*, encoding the major cardiac sodium channel, influence impulse conduction and are associated with a broad spectrum of arrhythmia disorders. Here, we identify an evolutionary conserved regulatory cluster with super enhancer characteristics downstream of *SCN5A*, which drives localized cardiac expression and contains conduction velocity-associated variants. We use genome editing to create a series of deletions in the mouse genome and show that the enhancer cluster controls the conformation of a >0.5 Mb genomic region harboring multiple interacting gene promoters and enhancers. We find that this cluster and its individual components are selectively required for cardiac *Scn5a* expression, normal cardiac conduction and normal embryonic development. Our studies reveal physiological roles of an enhancer cluster in the *SCN5A-SCN10A* locus, show that it controls the chromatin architecture of the locus and *Scn5a* expression, and suggest that genetic variants affecting its activity may influence cardiac function.

[1] Department of Medical Biology, Amsterdam Cardiovascular Sciences, Academic Medical Center, Amsterdam, The Netherlands. [2] Oncode Institute, Hubrecht Institute-KNAW and University Medical Center Utrecht, Utrecht, the Netherlands. [3] These authors contributed equally: Rajiv A. Mohan, Malou van den Boogaard. *email: v.m.christoffels@amc.uva.nl

The *SCN5A* gene encodes the alpha subunit of the major voltage-gated sodium channel Na$_v$1.5. Cardiac sodium channels are essential for fast conduction of electrical impulses through myocardium and as such play an important role in excitation and contraction of the heart. Deletions, and gain- or loss-of-function mutations in *SCN5A* are associated with a spectrum of human cardiac conduction diseases including bradycardia, atrial fibrillation, and Brugada syndrome. Only in about 25% of the Brugada syndrome patients, however, a mutation is found in the coding sequence of *SCN5A*, indicating a role of other genes or of other mechanisms affecting *SCN5A* expression[1,2]. Genome-wide association studies (GWAS) have identified genetic variants in the *SCN5A-SCN10A* locus associated with conduction velocity as indicated by the PR interval and QRS duration of the ECG[3–5]. Interestingly, an unexpectedly large cumulative effect of variants in the *SCN5A-SCN10A* and *HEY2* loci on Brugada syndrome susceptibility was discovered[6]. The majority of disease-associated genetic variants are found in non-coding regulatory DNA regions that control gene expression[7].

The spatial and temporal expression patterns of genes are controlled by regulatory elements (REs), which include enhancers[8]. REs are occupied by lineage-specific and general transcription factors and interact with promoters of their target genes in order to regulate transcription. Distinct epigenetic signatures have identified over a million of putative REs in different human and mouse cell types[9–11]. REs and their target genes are usually present within the same topologically associated domains (TADs)[8,12–14]. The expression of most genes is regulated by multiple REs, and REs may control multiple target genes. The regulatory relationships among active REs is complex (e.g. additive, synergistic), condition-specific (e.g. cell-type, developmental stage) and has not been defined for the vast majority of genes and conditions[8,15]. Particular epigenetic signatures and activities indicate the existence of large densely clustered REs, dubbed "super enhancers", which are often found near genes regulating cell identity[16,17]. Genes regulated by super enhancers are sensitive to perturbations leading to large phenotypic changes. While RE function has been extensively studied, insight into the in vivo physiological role and function of REs in mammals is limited[8,18].

Trait- or disease-associated variants supposedly alter transcription factor binding sites which changes the activity of REs, thereby modulate target gene expression[15]. Nevertheless, in vivo functional variants are rarely identified and the underlying mechanism usually remains obscure[19,20]. Moreover, because REs can be located kilobases (Kb) away from their target genes, ignoring genes and REs in between, it remains difficult to predict which genes are influenced by a particular variant RE. Locus-specific and genome-wide physical proximity maps have been generated through chromosome conformation capture technologies, which has provided valuable information regarding chromatin topology and possible interactions between putative regulatory sequences and promoters[12,21,22]. These maps are mostly derived from cultured, non-cardiac cells, and most datasets are of limited resolution. Recently, promoter-capture Hi-C maps of cardiomyocytes derived from human stem cells have been generated with greatly increased power to detect interactions involving promoter sequences in a cardiac relevant cell type[23,24]. However, close proximity by itself does not predict whether a regulatory sequence regulates a particular gene[14], and the number of in vivo proven target genes of tissue-specific REs in mammals is still limited.

Previously, we and others have identified several cardiac-specific REs in the human *SCN5A-SCN10A* locus that are capable of driving reporter gene expression in the embryonic mouse heart in patterns resembling that of *Scn5a*[4,25–27]. An RE in an intron of *SCN10A* (RE1) was found to harbor a common genetic variant associated with PR interval[5] that disrupts a T-box factor binding site, which was associated with decreased *SCN5A* expression levels in human hearts[27]. RE5 was identified in an intron of *SCN5A* and RE6 downstream of *SCN5A*. RE6 contains genetic variants associated with PR interval and QRS duration[3–5].

Here, we focus on the function of putative REs downstream of *SCN5A* that are part of a large cluster showing the hallmarks of a "super enhancer"[16,17]. Using genome editing technologies we delete different portions of the enhancer cluster from the mouse genome. We demonstrate that deletion of the enhancer cluster and its individual components cause a decrease in *Scn5a* expression and conduction slowing. Furthermore, deletion of the enhancer cluster causes a marked change in the chromatin architecture of the locus. Our studies reveal the regulatory function of an enhancer cluster in the *SCN5A-SCN10A* locus, and further suggest the relation between genetic variants found in non-coding regions in the *SCN5A-SCN10A* locus, RE function and the susceptibility to arrhythmias.

## Results

**Identification of a cardiac-specific *SCN5A* super enhancer**. To determine the TADs boundaries of the *SCN5A-SCN10A* locus, we used available Hi-C data derived from lymphoblastoid cell line GM12878[21,28]. TADs are largely invariant between different cell types and conserved across species[8,12]. We found that *SCN5A* and *SCN10A* share the TAD with *EXOG*, *SCN11A*, and *WDR48* (Fig. 1a). Enrichment of interactions within this region observed in promoter-capture Hi-C data of iPSC-derived cardiomyocytes[23] further supports the assignment of the TAD (Fig. 1a). The spatial organization of this TAD is conserved in between human (Fig. 1a) and mouse[27]. The activity of REs is largely limited to target genes that fall within the same TAD[8], including *SCN5A, SCN10A, EXOG, SCN11A*, and *WDR48*.

GWAS have identified genetic variants in the *SCN5A-SCN10A* locus that modulate ECG parameters and contribute to conduction diseases. We plotted the $-\log(P\ \text{value}) < 0.05$ of the genetic variants associated with PR interval and QRS duration[3–5] in the *SCN5A-SCN10A* locus. Interestingly, variants significantly associated with PR interval were mainly located over *SCN10A*, whereas variants associated with QRS duration in left ventricular hypertrophy were distributed over *SCN5A* and *SCN10A* (Fig. 1a). We observed co-localization of QRS duration-associated variants with increasing statistical significance in the *EXOG-SCN5A* intergenic region (Fig. 1a). Based on EMERGE prediction[29] (Supplementary Fig. 1 and Supplementary Table 1), the intergenic region is composed of four putative REs, which we named RE6-9 (Fig. 1b). Because of its size and its extensive association with H3K27ac[30], the entire region has been classified as a "super enhancer" in human and mouse[16,31]. Moreover, in human iPSC-derived cardiomyocytes, the region is occupied by Mediator 1, another hallmark of a super enhancer (Fig. 1b)[32]. We previously tested the activity of the human RE (hRE) 6, 7, and 9 independently in an Hsp68/LacZ enhancer reporter assay, which showed in vivo activity in different compartments of embryonic transgenic mouse hearts (Fig. 1c)[4,25]. Next, we examined conservation of the intergenic region RE6-9 between mouse and human. hRE6-9 covered 97.4% of span and 61.7% base pair alignment to the mouse genome. Based on a mouse cardiac-specific H3K27ac signature[33] and EMERGE enhancer prediction[29] (Supplementary Fig. 1 and Supplementary Table 1), the four REs are also identified in the *Exog-Scn5a* intergenic region (Fig. 2a, b). Furthermore, the mouse RE6-9 (mRE6-9) is occupied by various cardiac transcription factors (e.g. Tbx3, Tbx5, Tbx20, Nkx2-5 and Gata4) and other epigenetic motifs and signatures

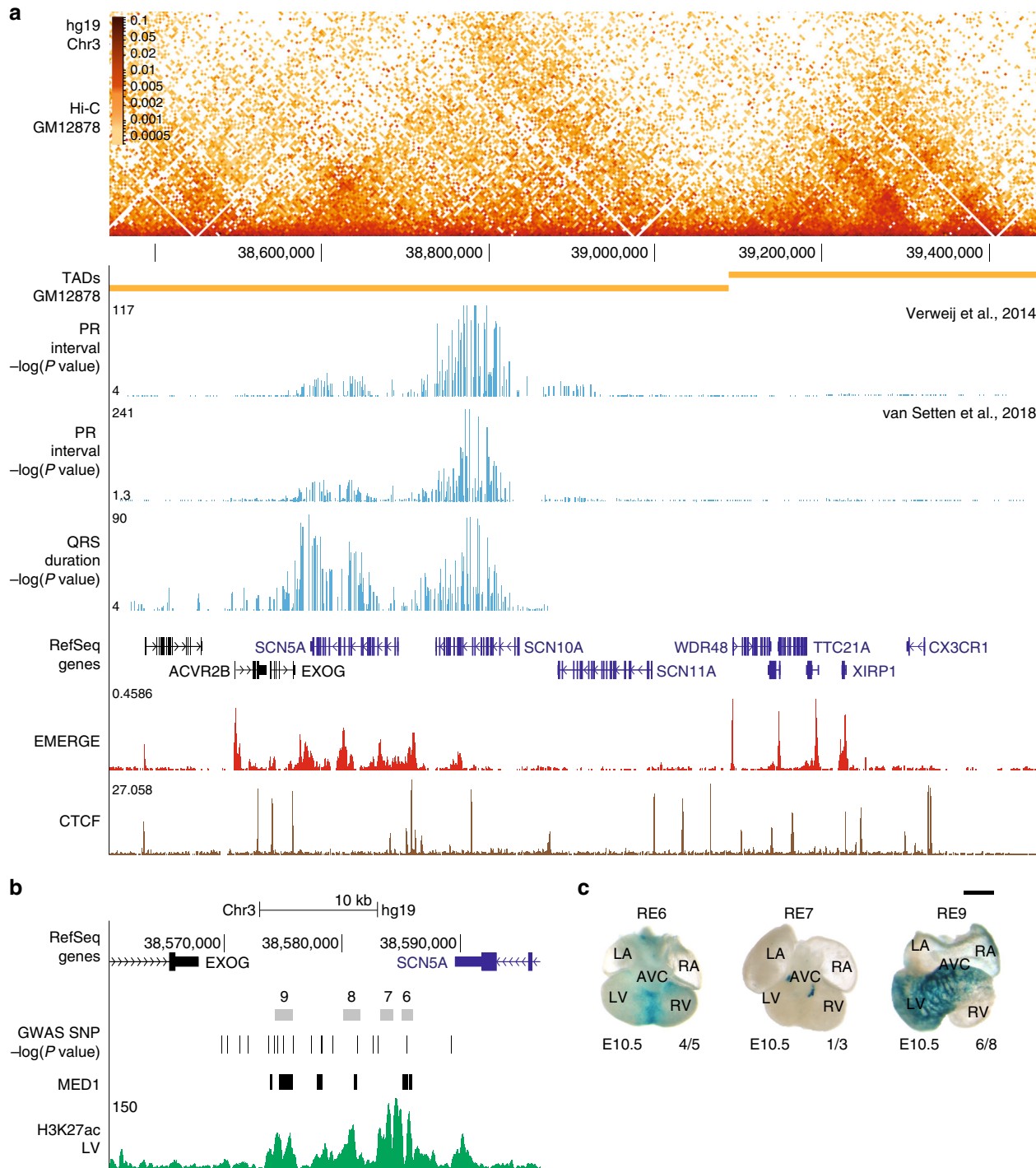

**Fig. 1** Identification of a cardiac-specific, multi-component 'super enhancer' in the human *SCN5A-SCN10A* locus. **a** Hi-C heatmap, visualized using HiGlass, from human lymphobastoid line GM12878[21] of the *SCN5A* TAD. -log(*P* values) of the genetic variants associated with PR interval[3,5] and QRS duration[4], human EMERGE track[29] and CTCF ChIP human cardiac muscle cells are included of the *SCN5A-SCN10A* locus. **b** UCSC browser view of the intergenic region between *SCN5A* and *EXOG*, composed of four constituent regulatory elements (RE6-9) with "super enhancer" characteristics based on human EMERGE, H3K27ac ChIP human left ventricle and Mediator 1 (MED1) ChIP from iPSC-derived cardiomyocytes[32]. PR interval-associated and QRS duration-associated (–log(*P* value)) variants are shown in the enhancer cluster. **c** Dorsal view of E10.5 hearts containing human regulatory elements RE6[4], 7 and 9[25] in a Hsp68/LacZ enhancer reporter vector. Human RE6 (4/5) is active in the interventricular septum and 9 (6/8) shows in vivo activity in the interventricular septum and left ventricle of embryonic mouse heart. RE7 (1/3) displays only cardiac activity in the embryonic outflow tract. Black scale bar is 0.5 mm. LA left atrium, LV left ventricle, RA right atrium, RV right ventricle, AVC anterior ventricular canal

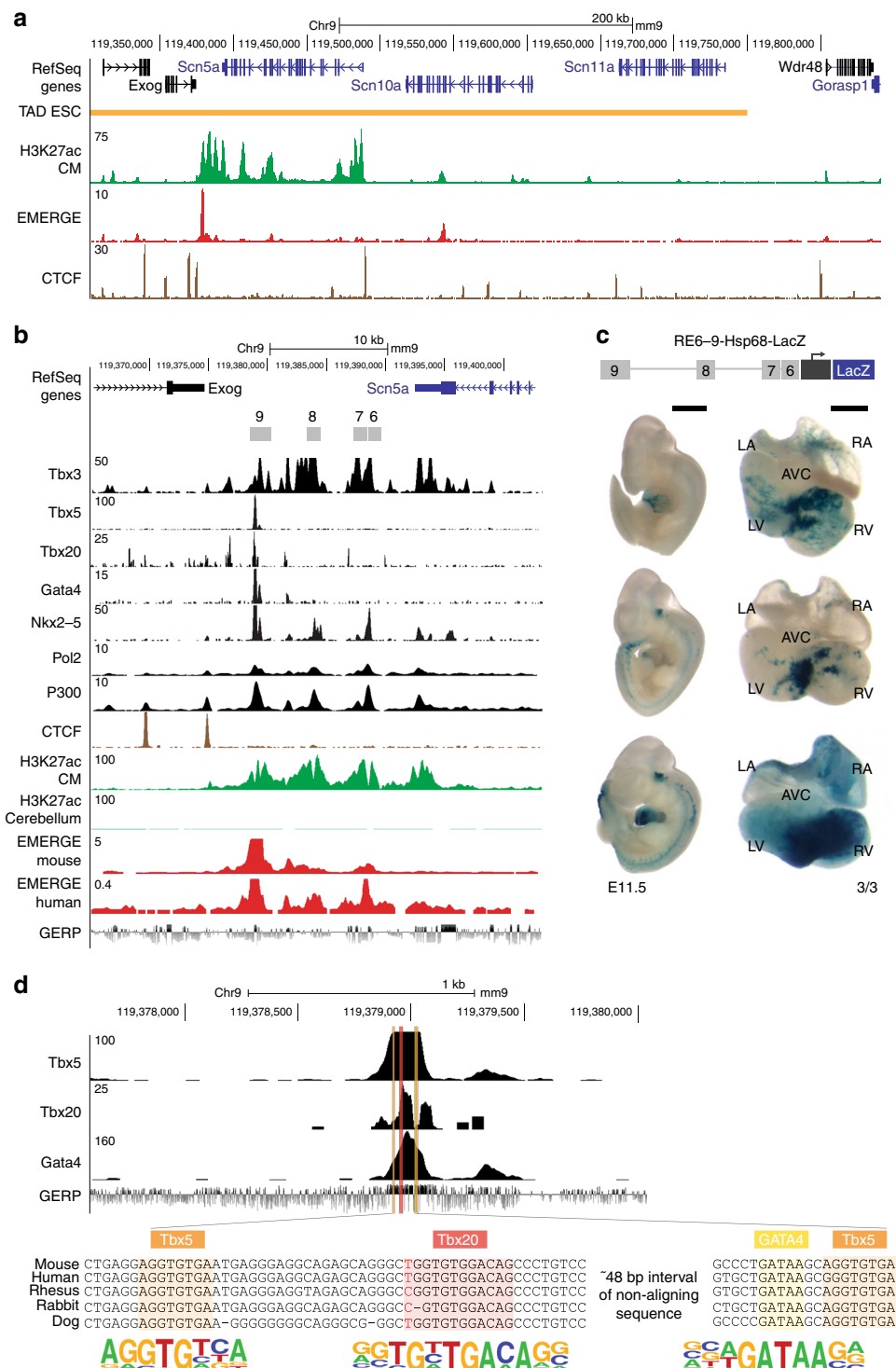

**Fig. 2** Identification of a cardiac-specific, multi-component 'super enhancer' in the mouse *Scn5a-Scn10a* locus. **a** UCSC browser view of *Scn5a-Scn10a* locus with TAD obtained from Hi-C data on mouse embryonic stem cells (ESC), cardiomyocyte-specific H3K27ac ChIP[33], mouse EMERGE[29] and CTCF ChIP adult mouse heart. **b** UCSC browser view of the intergenic region between *Scn5a* and *Exog*, composed of four constituent regulatory elements (RE6-9) with ChIP-seq profiles of Tbx3 (black), Tbx5 (black), Tbx20 (black), Gata4 (black), Nkx2-5 (black), Pol2 (black), P300 (black) from the heart. H3K27ac (green) in cardiomyocytes and cerebellum, mouse and human EMERGE[29] tracks (red) and conservation track GERP (bottom). **c** Dorsal view of mouse regulatory region RE6-9 in a Hsp68/LacZ enhancer reporter vector. Mouse RE6-9 (3/3) shows predominant activity in the interventricular septum and ventricles and to a lesser extent in the atria, in 2 of 3 E11.5 embryos. Black scale bars are 1 mm (embryo) and 0.5 mm (heart). **d** An example showing a zoomed-in view of predicted transcription factor binding sites Tbx5, Tbx20 and GATA4 in RE9 including sequence alignment between species and JASPAR consensus site. LA left atrium, LV left ventricle, RA right atrium, RV right ventricle, AVC anterior ventricular canal

(e.g. Pol2, H3K27ac and p300) (Fig. 2b, d). We functionally tested mRE6-9 in vivo. The mouse enhancer cluster was strongly active in the interventricular septum and the ventricles, and to a lesser extent in the atria. Activity was reproducibly absent from the atrioventricular canal and outflow tract (Fig. 2c). This pattern is reminiscent of that of endogenous Scn5a and Scn10a in the developing heart[25,34]. This demonstrates that RE6-9 is a cardiac-specific enhancer cluster and that its activity is evolutionary conserved between human and mouse.

**RE6-9 controls the chromatin architecture of the Scn5a-Scn10a locus.** The three-dimensional organization of the Scn5a-Scn10a locus suggests that the intergenic region RE6-9 downstream of Scn5a contacts several sites within this locus including the promoters of Exog, Scn5a, and Scn10a and several REs (Fig. 1a). This is further supported by ChIP-seq data of CCCTC-binding factor (CTCF). CTCF contributes to looping and interactions between promoters end REs, and to TAD boundaries[35–37]. We found that in both the human and mouse genome conserved converging CTCF-occupied sites flank RE6-9 and the Scn5a promoter, and RE1 and Scn11a, respectively (Fig. 2a and Supplementary Fig. 2). To explore the role of RE6-9 in the conformation of the locus, we deleted roughly 17 Kb of the intergenic region in which RE6-9 is located from the mouse genome using TALEN technology, without disrupting the 3′ untranslated regions (UTR) of Scn5a and Exog, respectively. We investigated the three-dimensional organization of the locus in whole hearts of 8-week-old heterozygous RE6-9 (RE6-9$^{+/-}$) mice ($n = 3$) versus wildtype littermates ($n = 3$) by high-resolution chromosome conformation capture-sequencing (4C-seq). Using the promoter region of Scn5a as viewpoint, we found that it contacts multiple sites within the TAD including the promoters of Scn10a and Exog, and RE6-9, RE5 in Scn5a[4], and RE1 in Scn10a[27] in both RE6-9$^{+/-}$ mice (orange signal) and wildtype littermates (blue signal) (Fig. 3a and Supplementary Fig. 3). The conformation of the locus resembled that observed in liver (Supplementary Fig. 4), indicating that the TAD is stable between different tissues. We tested for changes in average contact profiles between RE6-9$^{+/-}$ and wildtype mouse hearts for a set of positions in the Scn5a-Scn10a locus (Supplementary Fig. 5). A significant reduction ($P = 0.0009$) in interaction signal between the Scn5a promoter and RE6-9 was observed in the RE6-9$^{+/-}$ mice when compared to wildtype. Moreover, interactions with Scn10a promoter ($P = 0.02$), RE5 ($P = 0.01$), and RE1 ($P = 0.04$) were also reduced (Supplementary Fig. 5). In contrast, Exog, Wdr48 and genes outside the TAD showed no change in interactions with the Scn5a promoter region (Fig. 3a and Supplementary Fig. 5). Note the changes in interaction signal are limited because in the heterozygous RE6-9$^{+/-}$ mice a wild-type allele maintaining interactions is still present. Homozygous mutants were not investigated as they do not survive beyond embryonic stages (see below). Next, we examined the topology of the locus from a different perspective by setting the viewpoint at RE1. We confirmed that RE1 interacts with RE6-9 and Scn5a and Scn10a promoter regions (Fig. 3b and Supplementary Fig. 6)[25,27]. Contact frequency between RE1, RE6-9 ($P = 0.003$) and the promoters of Scn5a ($P = 0.001$) and Scn10a ($P = 0.003$) were reduced in ventricles of RE6-9$^{+/-}$ mice (Supplementary Fig. 7). Taken together, 4C-seq analysis revealed that the three-dimensional organization of the locus changes upon deletion of RE6-9, and that the intradomain interactions between Scn5a, Scn10a, RE6-9, RE5, and RE1 were reduced (Fig. 3c).

**RE6-9 strikingly selective controls Scn5a expression.** RE6-9 interacts with multiple promoters and is required for the topological organization of the TAD, indicating it may directly or indirectly control expression of multiple genes within the TAD. We assessed the transcript levels of Scn5a and genes in- and outside the TAD in 2-week-old ventricles and brain tissue of RE6-9$^{+/-}$ mice by RT-PCR (Fig. 4a, b). Scn5a was prominently affected in the heart, but not in the brain of RE6-9$^{+/-}$ mice compared to wildtype littermates. In addition, we observed a trend of decreasing Scn10a expression in the heart. The transcript levels of the selected genes surrounding the Scn5a-Scn10a locus were unaffected in the heart and brain (Fig. 4a, b). To validate that the enhancer cluster specifically regulates Scn5a expression in the heart, we performed RNA-sequencing of 2-week-old ventricles of wildtype ($n = 3$) and RE6-9$^{+/-}$ mice ($n = 3$). We detected approximately 15.000 transcripts in the ventricles and surprisingly, within the TAD only Scn5a expression was highly significant downregulated ($P = 3.75e-27$) in RE6-9$^{+/-}$ mice (Fig. 4c–f). We noted that there was a significant decrease in Rnu5g expression. Rnu5g is a 5S small nuclear RNA located about 50 megabases (Mb) away from Scn5a on the same chromosome. These data indicate that RE6-9 is a cardiac-specific enhancer cluster that selectively controls Scn5a.

The two-fold downregulation of Scn5a expression suggests that only the wildtype allele is actively transcribing the gene upon heterozygous deletion of RE6-9. To assess the allele-specific expression of Scn5a we compared the genome of FVB/NRj with the genome of other mouse strains for sequence variations using www.sanger.ac.uk/science/data/mouse-genomes-project. Only SPRET/EiJ contains sufficient sequence variation in Scn5a, Scn10a, and Exog to enable the analysis of allele-specific expression. We generated F1 hybrids of heterozygous RE6-9 transgenic mice (FVB/NRj) crossed to SPRET/Eij mice. RNA isolated from left ventricular tissue of 8-week-old SPRET;FVB F1 hybrids was used to amplify RT-PCR products containing a genetic variant (SNP) in Scn5a, Scn10a, and Exog followed by Sanger sequencing. In F1 hybrid mice with an unmodified FVB allele, both the SPRET and FVB alleles expressed Scn5a based on the presence of both variants in the RNA (A/G and T/C; Fig. 5b, upper panel). In contrast, SPRET;FVB$^{ΔRE6-9}$ F1 hybrid mice only express a transcript from the SPRET-derived allele (A/T; Fig. 5b, upper panel). These data indicate that heterozygous deletion of RE6-9 results in a monoallelic expression of Scn5a, supports the notion that the enhancer cluster is absolutely required for the expression of cardiac Scn5a from the same allele, and argues against inter-chromosomal interaction between the enhancer and Scn5a promoter. In addition, Scn10a and Exog are expressed at the same ratios between the two different alleles in the SPRET;FVB and SPRET;FVB$^{ΔRE6-9}$ F1 hybrids, indicating that any changes in expression of these genes when RE6-9 is deleted are small and not detectable using this assay (Fig. 5b, lower panel).

We further investigated the transcript level of Scn5a in the cardiac compartments of 8-week-old RE6-9$^{+/-}$ mice using quantitative RT-PCR. We observed a 50% reduction in Scn5a expression in different compartments of the heart, except the right atrium, of heterozygous RE6-9 deletion mice (Fig. 5c). A similar downregulation was observed in mice heterozygous for a TALEN-generated 7-bp deletion in coding exon 3 of Scn5a, creating a frameshift with a premature stop codon. This mutation probably results in nonsense mediated decay of the Scn5a mRNA. Heterozygous deletion of RE6-9 and of the 7-bp in Scn5a exon 3 (Scn5a$^{+/Δ7}$) resulted in a significant decrease in the Scn5a mRNA level and Na$_v$1.5 protein level in the left ventricle (Fig. 5d, e). These data demonstrate that RE6-9 is required for Scn5a expression in multiple parts of the heart. Loss-of-function gene mutations may trigger the induction of genes exhibiting sequence similarity with the mutated gene, a phenomenon called genetic compensation[38,39]. We measured the expression of sodium

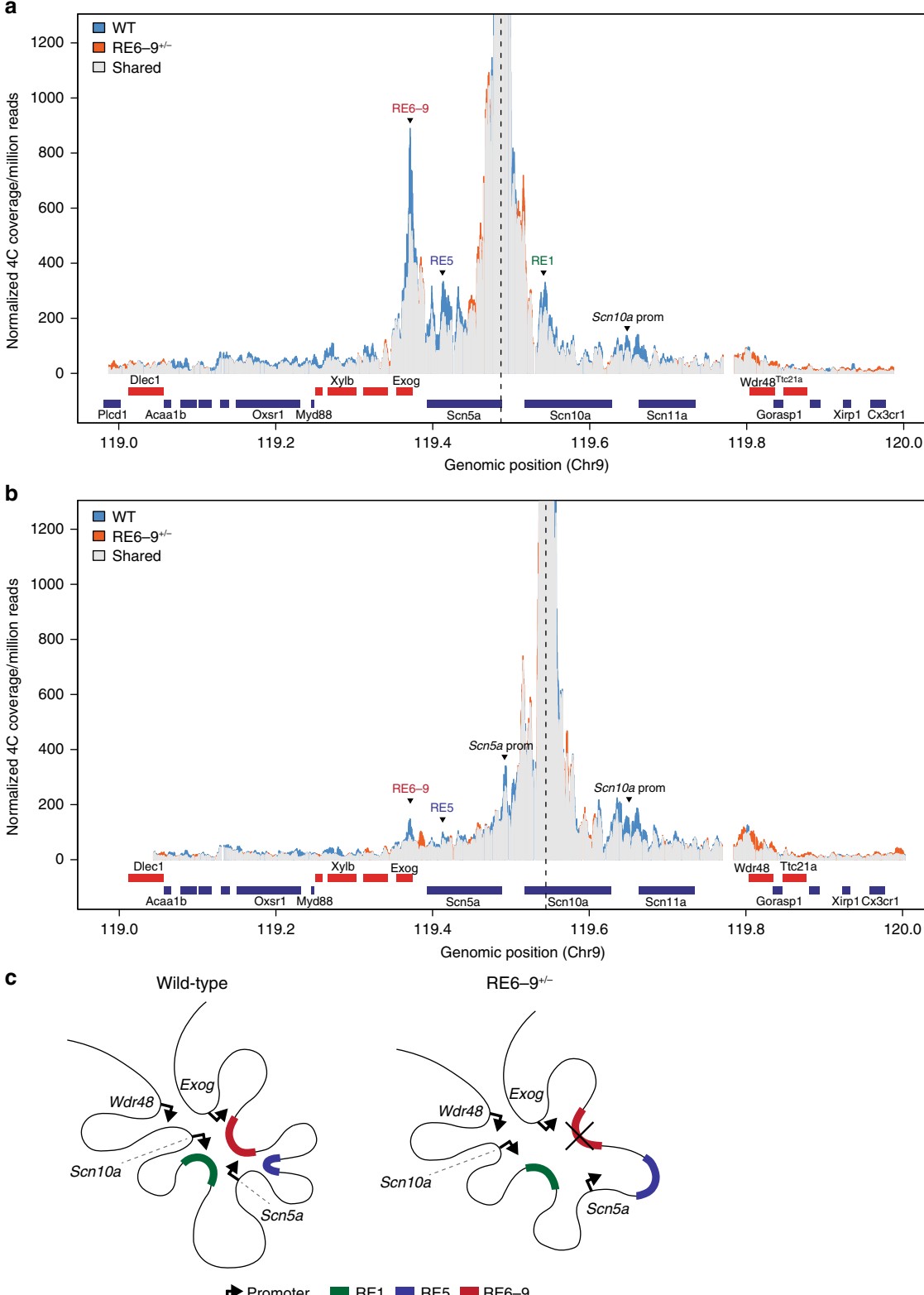

**Fig. 3** The three-dimensional organization of the *Scn5a-Scn10a* locus changes upon deletion of RE6-9. **a**, **b** 4C contact profiles of wildtype, $n = 3$ (WT, blue), RE6-9$^{+/-}$, $n = 3$ (orange) and shared signal (gray) with viewpoints set on the *Scn5a* promoter (**a**) and the intronic regulatory element RE1 located in *Scn10a* (**b**). The intradomain interactions between *Scn5a*, *Scn10a*, RE6-9, RE1, and RE5 are reduced in heterozygous RE6-9 mice. **c** Model of interaction of the *Scn5a-Scn10a* locus. Regulatory elements RE1 (green), RE5 (blue) and RE6-9 (red) with the *Scn5a*, *Scn10a*, *Exog*, and *Wdr48* promoters (black arrow)

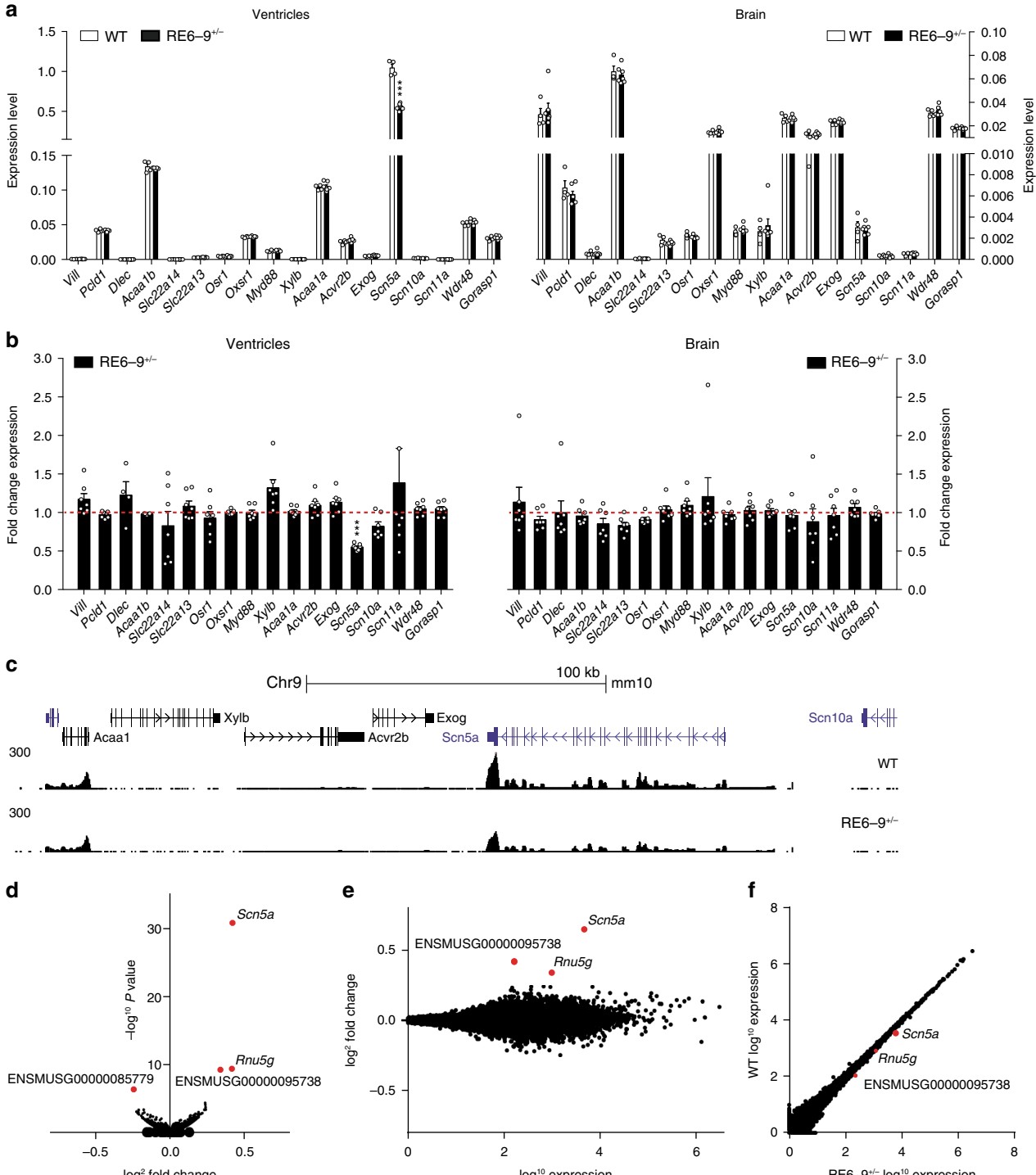

**Fig. 4** Transcriptome analysis of heterozygous RE6-9$^{+/-}$ mice. **a**, **b** Absolute expression (**a**) and fold change expression (**b**) levels of *Scn5a*, *Scn10a* and genes within and outside *Scn5a* TAD from ventricles (left) and brain (right) from 2-week old RE6-9$^{+/-}$ mice compared to wiltype (WT). Bars represent mean ± SEM and *P* values were calculated using unpaired Student's *t* test. WT n = 4; RE6-9$^{+/-}$ n = 7. ***P < 0.001. **c** UCSC browser view of the *Scn5a*-*Scn10a* locus showing RNA-seq profile of WT and RE6-9$^{+/-}$ mice. **d**–**f** Vulcano plot (**d**), MA plot (**e**) and scatter plot (**f**) of approximately 15.000 transcripts showing only *Scn5a* is highly significantly (*P* = 3.75e-27) downregulated in RE6-9$^{+/-}$ mice compared to WT. *P* values were corrected for multiple testing using the false discovery rate (FDR) method of Benjamini-Hochberg (*P* < 0.05). Source data are provided as a Source Data file

channel gene family members of *Scn5a*, and found deregulation of several members in the heart of heterozygous *Scn5a*$^{+/\Delta 7}$ mice (Supplementary Fig. 8). However, expression of these family members was not changed in RE6-9$^{+/-}$ mice, showing that reduced expression of *Scn5a* in RE6-9$^{+/-}$ mice by itself does not trigger genetic compensation (Supplementary Fig. 8).

**RE6-9 enhancer integrity is required for *Scn5a* expression**. To explore the functional integrity of the RE6-9 enhancer cluster depends on individual elements, we established transgenic mouse lines with deletions of elements utilizing CRISPR/Cas9 and TALEN technology (Fig. 5a). Using quantitative RT-PCR we determined *Scn5a* expression in the ventricles of 8-week-old

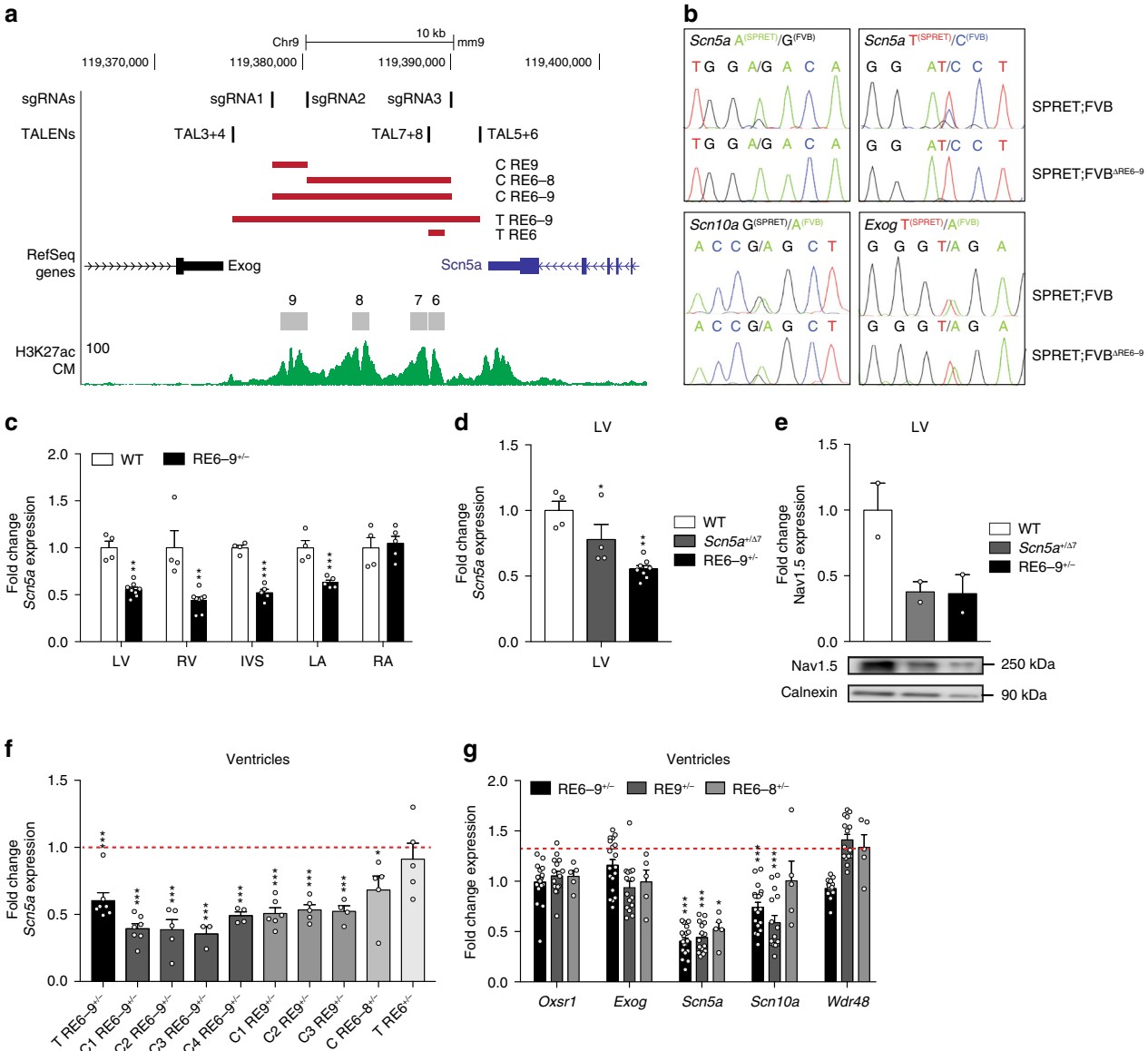

**Fig. 5** Integrity of the enhancer cluster RE6-9 is required for its function in the heart and cardiac *Scn5a* expression. **a** UCSC browser view of the enhancer cluster RE6-9 and the deletions of regulatory elements within this region by TALEN and CRISPR/Cas9 technology (red rectangles). Included location of the TALENs and CRISPR/Cas9 sgRNAs design and cardiomyocyte-specific H3K27ac ChIP[33]. **b** Sequence traces of a genetic variant in *Scn5a*, *Scn10a* and *Exog* from SPRET;FVB and SPRET;FVB$^{\Delta RE6-9}$ F1 hybrids. **c** Fold change expression level of *Scn5a* in different compartments of heart from 8-week old RE6-9$^{+/-}$ mice compared to wildtype (WT). *P* values were calculated using unpaired Student's *t* test. WT $n = 4$; RE6-9$^{+/-}$ $n = 5$ (LA, RA, IVS); RE6-9$^{+/-}$ $n = 7$ (LV, RV). **$P < 0.01$; ***$P < 0.001$. **d** Fold change expression level of *Scn5a* from left ventricle of 8-week old RE6-9$^{+/-}$ and *Scn5a*$^{+/\Delta 7}$ mice. *P* values were calculated using unpaired Student's *t* test. WT $n = 4$; *Scn5a*$^{+/\Delta 7}$ $n = 4$; RE6-9$^{+/-}$ $n = 5$. *$P < 0.05$; **$P < 0.01$. **e** Fold change expression level of Nav1.5 from left ventricle of 8-week old RE6-9$^{+/-}$ and *Scn5a*$^{+/\Delta 7}$ mice. WT $n = 2$; *Scn5a*$^{+/\Delta 7}$ $n = 2$; RE6-9$^{+/-}$ $n = 2$. **f** Fold change expression level of *Scn5a* from various established mouse lines (8-week old) with deletions of regulatory elements from the enhancer cluster. *P* values were calculated using unpaired Student's *t* test. T RE6-9$^{+/-}$ $n = 5$; C1 RE6-9$^{+/-}$ $n = 7$; C2 RE6-9$^{+/-}$ $n = 5$; C3 RE6-9$^{+/-}$ $n = 3$; C4 RE6-9$^{+/-}$ $n = 4$; C1 RE9$^{+/-}$ $n = 6$; C2 RE9$^{+/-}$ $n = 5$; C3 RE9$^{+/-}$ $n = 4$; C RE6-8$^{+/-}$ $n = 5$; T RE6$^{+/-}$ $n = 5$. *$P < 0.05$; ***$P < 0.001$. **g** Fold change expression level of *Oxsr1*, *Exog*, *Scn5a*, *Scn10a*, and *Wdr48* of CRISPR transgenic mouse lines RE6-9, RE9, or RE6-8. *P* values were calculated using unpaired Student's *t* test. *$P < 0.05$; **$P < 0.01$; ***$P < 0.001$. Bars represent mean ± SEM. LV left ventricle, RV right ventricle, IVS interventricular septum, LA left atrium, RA right atrium, T TALEN, C CRISPR/Cas9. Source data are provided as a Source Data file

heterozygous mutants in which individual or multiple elements of the enhancer cluster were deleted. Heterozygous deletion of RE6-8 (RE6-8$^{+/-}$) causes a 30% decrease in *Scn5a* expression, whereas heterozygous deletion of RE9 (RE9$^{+/-}$) caused a reduction of 50% in *Scn5a* expression, similar to the reduction in RE6-9$^{+/-}$ mice. In contrast, heterozygous deletion of RE6 (RE6$^{+/-}$), previously implicated in *SCN5A* expression and containing a GWAS variant[4], did not affect *Scn5a* expression in the ventricles (Fig. 5f).

Next, we assessed the transcript levels of *Oxsr1*, *Exog*, *Scn10a*, and *Wdr48* surrounding the *Scn5a* gene within the TAD in the heterozygous RE6-9$^{+/-}$, RE9$^{+/-}$, and RE6-8$^{+/-}$ mice. Interestingly, we observed a mild decrease in *Scn10a* and *Wdr48* expression after deletion of individual components of the enhancer cluster, whereas *Oxsr1* and *Exog* were unaffected (Fig. 5g). Together, these data indicate that deletion of individual components of the enhancer cluster RE6-9 causes loss of functional integrity of the

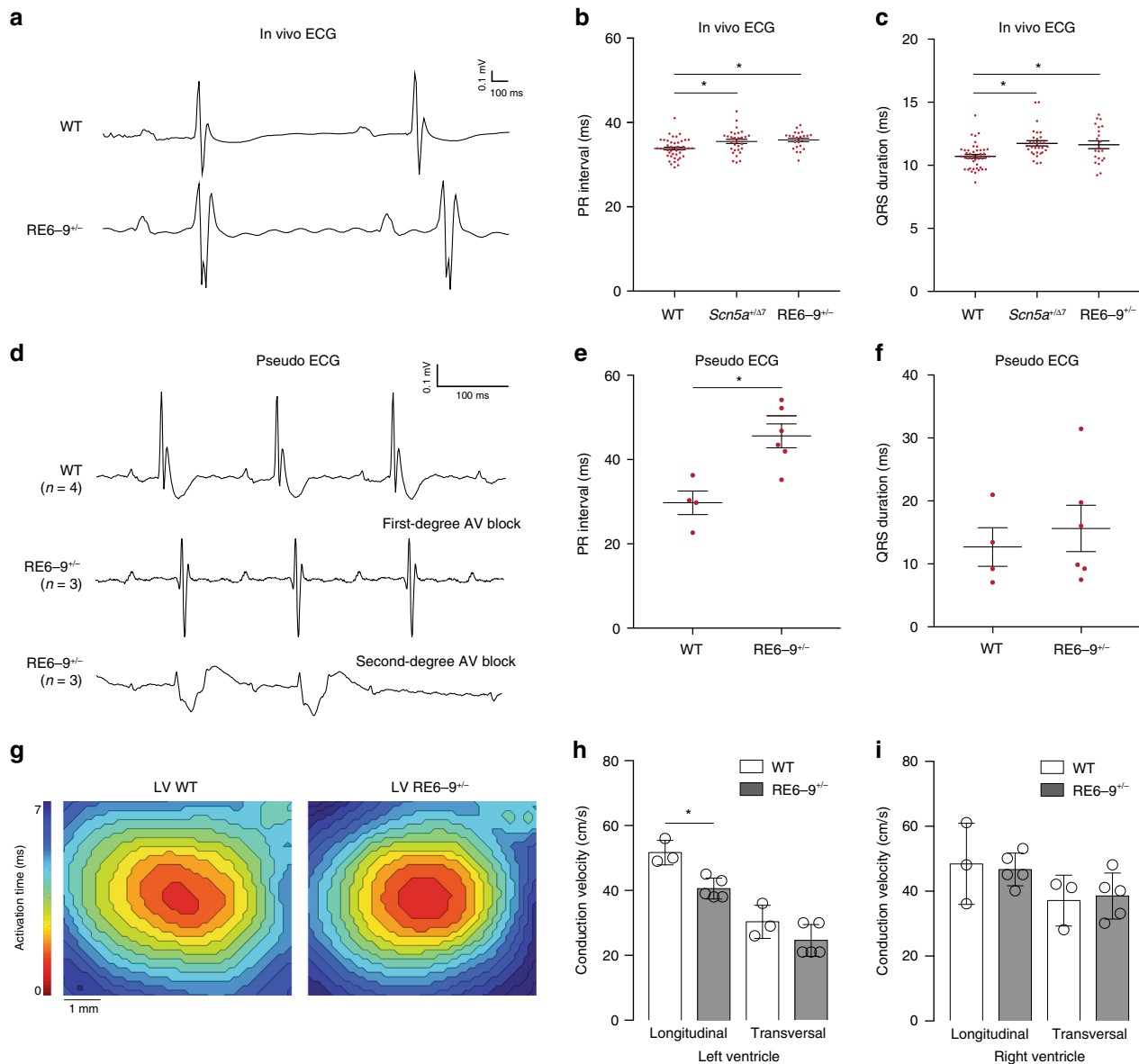

**Fig. 6** Deletion of the enhancer cluster RE6-9 causes conduction slowing. **a** Electrograms recorded in vivo from a wildtype (WT) and RE6-9$^{+/-}$ mouse. **b**, **c** Scatter graph showing values for PR interval (**b**) and QRS duration (**c**) in vivo in 8 to 13-week old WT, Scn5a$^{+/\Delta7}$, and RE6-9$^{+/-}$ mice. P values were calculated using one-way ANOVA with post-hoc Bonferroni tests. Left panel (**b**): WT $n = 45$; Scn5a$^{+/\Delta7}$ $n = 32$; RE6-9$^{+/-}$ $n = 23$, right panel (**c**): WT $n = 44$; Scn5a$^{+/\Delta7}$ $n = 31$; RE6-9$^{+/-}$ $n = 22$. *$P < 0.05$. **d** An example of electrograms recorded from Langendorff-perfused hearts from a WT and RE6-9$^{+/-}$ mouse with first-degree AV block and RE6-9$^{+/-}$ with second-degree AV block (Wenkebach). **e**, **f** Scatter graph showing values for PR interval (**e**) and QRS duration (**f**) ex vivo in 8 to 13-week old WT and RE6-9$^{+/-}$ mice. P values were calculated using unpaired Student's t test. WT $n = 4$; RE6-9$^{+/-}$ $n = 6$. *$P < 0.05$. **g** Reconstructed activation patterns of the left ventricle from a WT and RE6-9$^{+/-}$ mouse. **h**, **i** Left (**h**) and right (**i**) ventricular conduction velocity at longitudinal and transversal pacing in 8 to 13-week old WT and RE6-9$^{+/-}$ mice. P values were calculated using unpaired Student's t test. WT $n = 3$; RE6-9$^{+/-}$ $n = 5$. *$P < 0.05$. Bars represent mean ± SEM. Source data are provided as a Source Data file

cluster and subsequently loss of Scn5a expression, and point towards the prominent contribution of RE9 to enhancer cluster activity.

**Deletion of enhancer cluster RE6-9 causes conduction slowing.** Heterozygous Scn5a mutant mice display a variety of arrhythmias and progressive cardiac conduction disease including prolongation of the PR interval and QRS duration[40]. We recorded in vivo and ex vivo ECG, and performed optimal mapping from 8 to 13-week-old RE6-9$^{+/-}$ mice. ECG recordings indicated heart rates in anesthetized RE6-9$^{+/-}$ mice and wildtype littermates was not different. The ECGs of RE6-9$^{+/-}$ mice revealed a lengthening of

the PR interval (wildtype $n = 45$; RE6-9$^{+/-}$ $n = 23$) and QRS duration (wildtype $n = 44$; RE6-9$^{+/-}$ $n = 22$), whereas other ECG parameters were not altered (Fig. 6a–c). Next, we studied the electrophysiological properties of whole isolated RE6-9$^{+/-}$ hearts in a Langendorff perfusion setup. We observed prolongation of the PR interval in RE6-9$^{+/-}$ hearts ($n = 6$) compared to wildtype hearts ($n = 3$) (Fig. 6d–f). Furthermore, episodes of second-degree atrioventricular conduction (AV) block was observed in 3 of 6 RE6-9$^{+/-}$ hearts. Note the autonomic modulation is absent ex vivo, which is of crucial importance for AV conduction in the setting of reduced sodium current or sodium channel dysfunction. We speculate the autonomic balance maintains safety of AV conduction in affected AV nodes of RE6-9$^{+/-}$ mice in vivo,

providing an explanation for the differences in PR prolongation in in vivo and ex vivo ECG recordings.

We measured ventricular conduction velocity of RE6-9$^{+/−}$ mice during central stimulation. Longitudinal conduction velocity was decreased in the left ventricle of RE6-9$^{+/−}$ mice ($n = 5$) compared to wildtypes ($n = 3$) (Fig. 6g, h). The latter supports the observed QRS prolongation in vivo ECGs from RE6-9$^{+/−}$ mice (Fig. 6c). The conduction velocity was not lower in the right ventricle (Fig. 6i). Together, these results show that the enhancer cluster is required for electrophysiological homeostasis of the heart.

Although RE6$^{+/−}$ mice did not show altered *Scn5a* expression, we observed ~30% reduction in *Scn5a* expression in the left ventricle of homozygous RE6 (RE6$^{−/−}$) mice (Supplementary Fig. 9). Note, we removed the core of RE6 including the T-box site the homolog of which in human harbors rs6781009[4]. We investigated the functional role of RE6 in myocardial excitability and cardiac conduction by performing in vivo ECG analysis. ECG recordings from 8 to 13-week-old RE6$^{+/−}$ and RE6$^{−/−}$ mice indicated no functional effect on conduction in the adult mouse heart (Supplementary Fig. 9). Moreover, when we intercrossed heterozygous RE6 mutants in a sensitized genetic background carrying an *Scn5a* loss-of-function allele (*Scn5a*$^{+/Δ7}$), we did not observed an additional or cumulative effect on change in PR interval and QRS duration (Supplementary Fig. 9). These findings indicate that the contribution of RE6 in the regulation of *Scn5a* is minimal.

**RE6-9 is partially redundant during development**. We next analyzed the requirement of the enhancer cluster for development. Homozygous *Scn5a*$^{Δ7/Δ7}$ embryos died around E10.5, as was previously observed for *Scn5a* homozygous loss-of-function mice[40] (Fig. 7a). To assess the regulatory function of RE6-9 during development, we generated compound heterozygous mutants by interbreeding of *Scn5a*$^{+/Δ7}$ and RE6-9$^{+/−}$. *Scn5a*$^{Δ7/}$ $^{ΔRE6-9}$ embryos are also lethal at E10.5 and these embryos are phenotypically similar to *Scn5a*$^{Δ7/Δ7}$ embryos, revealing that the expression level of *Scn5a* from the ΔRE6-9 allele is very low during early development (Fig. 7a). RE6-9 knockout mutants (RE6-9$^{−/−}$) survive up to E13.5 (Fig. 7b and Supplementary Table 2). We observed that RE6-9$^{−/−}$ E10.5 embryos still express low levels of *Scn5a* in the heart (Fig. 7d, f). Furthermore, there was a significant decrease of *Scn10a*, but no changes in the expression of *Exog* and *Wdr48* (Fig. 7d, f). Inspection of H3K27ac ChIP-seq data at different stages of pre- and postnatal heart development[41] revealed RE6-9 is increasingly enriched for H3K27ac during development (Fig. 7h). These findings suggest that prior to the intermediate embryonic stage (E13.5) other REs (RE1, RE5) act during development to control *Scn5a* expression in the heart. Thus the enhancer cluster RE6-9 is partially redundant during development. Next, we tested the embryonic lethality in RE6-8 knockout mutants (RE6-8$^{−/−}$). RE6-8$^{−/−}$ embryos were normal compared to wildtype at E13.5 and we did not observe cardiac defects (Fig. 7c, Supplementary Table 2). Furthermore, the expression of *Scn5a*, *Scn10a*, *Exog,* and *Wdr48* were not affected in RE6-8$^{−/−}$ embryos (Fig. 7e, g). These findings suggest that upon deletion of RE6-8 other REs (RE9, RE1, and RE5) operate to regulate *Scn5a* expression during heart development.

**Genetic variants affecting the enhancer activity of RE6-9**. A number of genetic variants associated with prolonged PR interval and QRS duration[3–5] map to the RE6-9 cluster (Fig. 1a, b). Because RE9 is a key element within the cluster, we focused on QRS-associated genetic variant rs6810361, positioned in RE9

(Supplementary Fig. 10). RE9 overlaps with a site occupied by T-box transcription factors (Tbx5, Tbx3, Tbx20) as indicated by cardiac ChIP-seq data[25,42,43]. We identified a conserved Tbx20 binding consensus sequence in RE9. However, rs6810361 is located at the edge of the T-box binding site (Supplementary Fig. 10). We performed functional analysis on the major allele (i.e., T) and minor allele (i.e., C) of rs6810361 in RE9 in rat heart derived H10 cells. Stimulation with only the cardiac transcription factors Nkx2-5 and Gata4 showed a strong reduction in RE9 enhancer activity for the minor allele in H10 cells. This effect is possibly rescued by Tbx20-mediated regulation as we observed an increase in enhancer activity of RE9 (Supplementary Fig. 10). We observed no difference in activation by Tbx3 and Tbx5 for both alleles (Supplementary Fig. 10). These findings demonstrate that a single nucleotide alteration that is not located in the core element of a T-box transcription factor binding site might affect binding of other cardiac transcription factors, which results in a change in enhancer activity in vitro.

## Discussion

In this study, we investigated the physiological role of REs in the SCN5A-SCN10A locus. We identified an evolutionary conserved cardiac enhancer cluster in the locus that controls expression of the SCN5A gene during development and homeostasis. Deletion of the enhancer cluster or its component REs resulted in the loss of *Scn5a* expression from the modified allele, revealing the REs are absolutely required for *Scn5a* gene expression in an allele-specific manner. As a consequence, the enhancer cluster is required for embryonic development and to maintain normal conduction velocities in the heart.

Our expression profiling showed the REs are strikingly selectively required for *Scn5a* expression, with the expression of only *Scn10a* being partially dependent on the enhancer cluster or one of its components, RE9. Loss-of-function mutations in SCN5A affects conduction and heart function, and cause arrhythmias under particular circumstances. Moreover, homozygous loss of function is embryonic lethal, independent of the lost sodium current in fish[44], suggesting SCN5A has additional functions. Therefore, we expected to observe a large number of genes to be indirectly deregulated in adult mutant mouse hearts. Apparently, a 50% reduction in mRNA and protein expression causing conduction slowing does not affect homeostatic gene expression. Recent studies have uncovered a mechanism for genetic compensation, which can occur in response to loss-of-function gene mutations[38,39]. A key aspect of this mechanism is the requirement of mutant mRNA degradation for the induction of genes with sequence similarity (e.g. paralogous genes). Consistent with these findings, we observed that the *Scn5a* 7-bp deletion allele, which expresses a mutated mRNA that presumably is degraded, caused deregulation (both induction and reduction) of expression of several paralogous sodium channel-encoding genes, whereas the deleted RE6-9 allele, which is not transcribed, did not. Adaptation has been proposed to provide a mechanism for robustness; alleles that do not transcribe the mutated gene do not show adaptation and may give rise to more severe phenotypes[38]. Our data imply that loss-of-function mutations in SCN5A coding sequences may cause deregulation of other genes, possibly influencing cardiac function, whereas mutations or variants that reduce SCN5A transcription, such as those affecting RE9, will not cause gene deregulation. These data have implications for our understanding of conduction disease-causing mutations affecting SCN5A. It will be interesting to analyze the mutation-induced genetic adaptation in greater detail, and assess how it influences cardiac function and homeostasis.

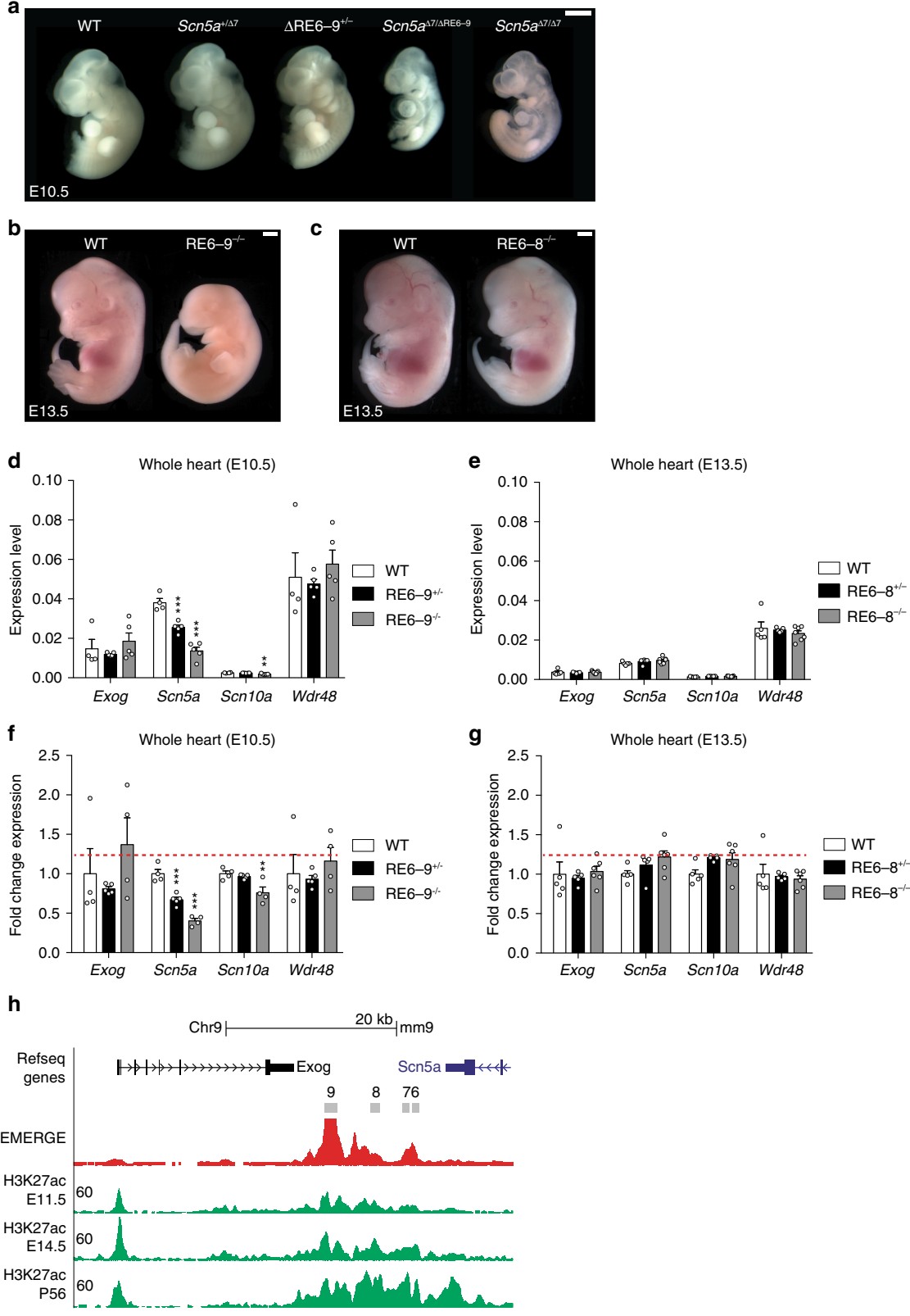

REs are usually located within the non-coding genome and often at large distances from gene promoters[8,14]. REs can stimulate transcription initiation by interacting with promoters via chromatin looping mechanisms. RE-promoter interactions are largely confined to TADs[13,36]. RE-promoter interactions have been mapped in various tissue types[45–47]. However, physical proximity between REs and promoters can be transient, tissue-specific and, importantly, does not predict whether the interaction is functional and influences transcription. Therefore, it remains difficult to predict the target promoter(s) of REs in a particular cell type or condition. We deleted key REs downstream of *Scn5a* from the mouse genome to define their target genes, tissue-specific functional interactions and mechanism of action. 4C-seq analysis indicated a change in the three-dimensional

**Fig. 7** Regulatory function of RE6-9 is partially compensated during development. **a** Analysis of $Scn5a^{+/\Delta 7}$ and RE6-9$^{+/-}$ embryos at E10.5. Heterozygous *Scn5a* and RE6-9 mutants develop normally. Compound heterozygous mutants $Scn5a^{\Delta 7/\Delta RE6-9}$ are lethal at E10.5 and are phenotypically similar to $Scn5a^{\Delta 7/\Delta 7}$. **b** RE6-9$^{-/-}$ embryos at E13.5 reveals embryonic lethality. **c** RE6-8$^{-/-}$ embryos at E13.5 develop normal and do not show cardiac defects. White scale bars are all 1 mm. **d, e** Absolute expression level of *Exog*, *Scn5a*, *Scn10a*, and *Wdr48* of E10.5 whole heart RE6-9$^{-/-}$ embryos (**d**) and E13.5 whole heart RE6-8$^{-/-}$ embryos (**e**) compared to wildtype (WT). **f, g** Fold change expression level of *Exog*, *Scn5a*, *Scn10a*, and *Wdr48* of E10.5 whole heart RE6-9$^{-/-}$ embryos (**f**) and E13.5 whole heart RE6-8$^{-/-}$ embryos (**g**). Bars represent mean ± SEM and *P* values were calculated using one-way ANOVA with post hoc Bonferroni tests. WT $n = 4$; RE6-9$^{+/-}$ $n = 5$; RE6-9$^{-/-}$ $n = 5$; WT $n = 5$; RE6-8$^{+/-}$ $n = 5$; RE6-8$^{-/-}$ $n = 7$. **$P < 0.01$, ***$P < 0.001$. **h** UCSC browser view of the enhancer cluster RE6-9 showing H3K27ac ChIP[41] at different stage of pre- and postnatal heart. Source data are provided as a Source Data file

organization of the *Scn5a* TAD in heterozygous RE6-9 mutants, involving a (partial) loss of interactions between several REs and promoters within the TAD. Multiple CTCF-occupied sites are present in this TAD including in the vicinity of RE6-9, the *Scn5a* promoter, RE1 and upstream of *Scn11a*. The RE6-9 deletion did not remove any CTCF-occupied site and indeed the 4C-seq profiles indicate the TAD boundaries were largely maintained. Despite the reduced interactions within the TAD, expression of only *Scn5a*, and to a much lesser extent *Scn10a*, was affected, indicating the enhancer cluster serves to set up or maintain the RE-promoter interactions of the region around *Scn5a-Scn10a*.

The RE6-9 enhancer cluster is large (>12 Kb), occupied by a high density of interacting factors including key cardiac lineage-specific transcription factors and Mediator 1[32], and strongly marked by cardiac-specific H3K27ac[30]. Furthermore, genetic variants associated with conduction traits have been identified that are localized in the enhancer cluster[3–5]. Together, these features classify the enhancer cluster as a "super enhancer" in the human and mouse genome[16,17,31]. Super enhancers are commonly found near and controlling genes important for defining cell identity[16,17]. Although the RE6-9 cluster of *SCN5A* seems not to be involved in the regulation of genes that control cell fate determination, it is strictly required for expression of *SCN5A*, a cardiomyocyte-specific gene essential for heart function and survival[1,2].

REs such as enhancers are classically described to act autonomous, modular, and additive; the activity of each RE in a (conserved) regulatory region adds its spatio-temporal activity pattern to that of the target gene(s)[8,15,48]. However, REs can act synergistic, repressive, competitive, or hierarchical (reviewed in ref. [8,15]). Super enhancers have been ascribed synergistic and hierarchical activities of its component REs[49,50]. However, the change in transcriptional output in response to deletion of super enhancer components and REs in mouse in vivo could also be explained by a linear model[48], the component REs acting in an additive manner[49,51–53]. Therefore, it is noteworthy that the RE6-9 cluster components, while able to activate transcription in the heart, show non-additive and hierarchical behavior when deleted. Deletion of RE9 was sufficient to abolish activity, deletion of RE6-8 reduced activity and deletion of RE6 caused a very mild reduction in activity.

Redundant enhancers (also known as "shadow" enhancers) share similar activity patterns or functions near the same gene[8,15,54]. They have been mostly studied in invertebrates, and have been proposed to provide an evolutionary conserved mechanism to ensure stable and precise gene expression patterning during development and stress responses[55]. Several studies of specific individual loci have identified mammalian REs near the same gene showing highly similar activity patterns[49,52,53,56]. Such REs in the mouse genome may act redundantly in vivo; removal of individual REs from the genome caused only slightly reduced expression of the target gene that does not result in phenotypic changes[53,57,58]. Four REs were identified in the *SCN5A-SCN10A* locus that have activity patterns

similar to the expression pattern of *SCN5A*[4,25]. *SCN5A* is dose sensitive; reduction in expression or function causes cardiac functional abnormalities and arrhythmias[6]. Therefore, *SCN5A* and the four REs form a regulatory system reminiscent of the buffered system described above. However, removal of the enhancer cluster (RE6-9) or RE9 resulted in complete loss of *Scn5a* expression from the modified allele, indicating the other REs (RE1, RE6, RE5) are not capable of providing redundancy. We speculate that RE5, RE6, and RE1 are not able to compensate because they need RE6-9 to maintain proximity to the *Scn5a* promoter. This might be different for the right atrium as we observed a lack of response in *Scn5a* expression. It could be that other REs, or the promoter of *Scn5a* act dominantly or redundantly in the right atrium. It will be interesting to test whether removal of RE5 or RE1 results in a similar loss of activity, or whether the entire system is hierarchically organized and depends on the enhancer cluster RE6-9. This also implies that *SCN5A* expression is exquisitely sensitive to genetic variations and mutations in the RE6-9 region. We provide one example of a common variant present in RE9 that alters its activity. In a systematic in vitro activity screen, several variants in the *SCN5A* locus were found to affect enhancer activity, including the variant in RE9[59]. These data confirm our findings, and further suggest the potential sensitivity of *SCN5A* gene regulation to common variants. Further studies are required to assess whether the variant in RE9 affects *SCN5A* expression in the human heart.

## Methods
**Ethics statement**. Animal care and experiments conform to the Directive 2010/63/EU of the European Parliament. All animal work was approved by the Animal Experimental Committee of the Academic Medical Center, Amsterdam, and was carried out in compliance with the Dutch government guidelines. All animal experiments were performed under DAE285.

**UCSC Genome Browser**. Human coordinates of the enhancer cluster RE6-9 was converted to the mouse genome in UCSC Genome Browser using liftOver[60].

**TALEN-mediated genome editing**. TALENs were designed with TAL Effector Nucleotide Targeter[61] to target *Scn5a* exon 3, RE6 and RE6-9. TALEN recognition site, space region and off-target effects of the designed TALEN pairs are described in Supplementary Table 3. TALENS were assembled according to the Golden Gate cloning protocol[61] and were in vitro transcribed by mMessage mMachine T7 Transcription Kit (Thermo Fisher Scientific). The TALEN mRNA (7.5-10 ng) was microinjected into the cytoplasm of one-cell embryo's for generation of founder mouse lines.

**CRISPR/Cas9 genome editing**. Three single guide RNA (sgRNA) constructs were designed to target the enhancer cluster RE6-9 downstream of *Scn5a* using the online tool ZiFiT Targeter[62]. The sgRNA sequences are summarized in Supplementary Table 3. The sgRNA constructs and Cas9 construct[63] were in vitro transcribed by the MEGAshortscript T7 Transcription Kit and the mMessage mMachine T7 Transcription Kit (ThermoFisher Scientific). The sgRNAs (10 ng/μl per sgRNA) and Cas9 mRNA (25 ng/μl) were microinjected into the cytoplasm of one-cell embryo's for generation of founder mouse lines.

**4C-sequencing**. 4C templates were prepared as previously described[64]. In short, snap frozen adult mouse hearts of *Scn5a* RE6-9 mutant mice and wildtype littermates were crushed, dissociated and dounce homogenized to obtain single cell

suspension. Chromatin was cross-linked with 2% formaldehyde in PBS with 10% FCS for 10 min at room temperature. Nuclei were isolated and cross-linked DNA was digested with DpnII, followed by proximity ligation. Cross-links were removed and a secondary restriction digestion with Csp6I performed, followed again by proximity ligation. Per viewpoint, 8 PCR reactions were performed on 200 ng of 4C template each, to a total of 1.6 μg. PCR reactions were pooled and purified for next generation sequencing. The 4C-seq PCRs were essentially performed as previously described[64] using reading primers (Scn5a_P1 + 2—GTCCCAAGGGCA-CACTGATC and Scn5a_enh1—GAGACCCACACGTTAGGATC) as 4C view-points combined with non-reading primers (Scn5a_P1 + 2—CCTCTAGAGAGCCTAGTCCC and Scn5a_enh1—ACGGTGAGGACAACATA-GAC). The primers were extended with Illumina adapter sequences. Sequencing reads were demultiplexed and subsequently trimmed using FourCSeq package python tool "demultiplex.py"[65]. Trimmed reads were aligned with bowtie2 (v2.2.5) using standard parameters and quality filter set to 1 (-q 1) and processed as described[64]. In short, reads are mapped to a restricted mouse reference genome (mm9) consisting of sequences directly flanking the 4C primary restriction enzyme sites (DpnII), termed 4C frag-ends. Non-unique frag-ends are discarded for posterior analysis. The frag-end with highest coverage is removed from the dataset in the normalization process and data are read-depth normalized to 1 million aligned intrachromosomal reads. 4C-Seq coverage profile is obtained using "running means", i.e. coverage averages of 21 consecutive 4C frag-ends.

**RNA-sequencing**. Total RNA was isolated from the ventricular tissue of heterozygous RE6-9 mice and wildtype littermates using Tri Reagent (Sigma Aldrich) according to the manufacturer's protocol (Sigma Aldrich) and further purified using Nucleospin RNA Clean-up Kit with DnaseI treatment (Macherey-Nagel). cDNA was prepared with the Truseq Stranded Total RNA Library Preparation Kit (Illumina, Part# 15031048 Rev. E). Total RNA (1 μg) was purified and sheared into small fragments followed by cDNA synthesis and ligation of the adapter. The quality and size distribution of the cDNA library templates were validated with Agilent DNA 1000 on 2100 Bioanalyzer (Agilent Technologies). cDNA samples were pooled per lane of paired-end 125 bp sequencing on an Illumina HiSeq 2500 instrument.

**Differential expression analysis**. Reads were mapped to mm10 build of the mouse transcriptome using STAR[66]. Differential expression analysis was performed using the DESeq2 package based on a negative binomial distribution model[67].

**RNA isolation and RT-qPCR**. Total RNA was isolated from cardiac and brain tissue using Tri Reagent (Sigma Aldrich) according to the manufacturer's protocol (Sigma Aldrich) and treated with Dnase I (Invitrogen). cDNA was reverse transcribed from 500 ng total RNA with oligo-dT primers (125 μM) and the Superscript II system (Invitrogen). Expression of different genes was assayed with quantitative RT-PCR using the LightCycler 480 Real-Time PCR system (Roche Diagnostics). The amplification protocol consisted of 5 min 95 °C followed by 45 cycles of 10 s 95 °C, 20 s 60 °C and 20 s 72 °C. Relative start concentration was calculated using LinRegPCR[68]. Values were normalized to the geomean of *cTnI* and *Hprt* (adult heart), *Eef1e1* (brain) and *Tnnt2* (embryonic heart) expression levels. A complete list of all RT-qPCR primers used in this study are listed in Supplementary Table 4.

**Western blot**. Total membrane fraction was isolated from left ventricular tissue using ice-cold lysis buffer (5 mmol/L, Tris-HCl pH 7.5, 2 mmol/L EDTA) supplemented with a protease inhibitor cocktail tablet (Complete Mini, Roch Diagnostics) and then centrifuged at 380*g* at 4 °C to remove cell debris. Whole cell lysate was centrifugated at 18620*g* for 30 min. The pellet containing the total membrane fraction was resuspended resuspension buffer (75 mmol/L Tris-HCl pH 7.5, 5 mmol/L EDTA, 12.5 mmol/L MgCl₂) supplemented with a protease inhibitor cocktail tablet. Protein concentrations were determined using the Protein Assay Reagent BCA kit (Thermo Fisher Scientific). Protein samples (50 μg) of the total membrane fractions were heated to 95 °C for 5 min in Laemmli buffer (5x Laemmle buffer: 300 mmol/L Tris-HCl pH 6.8, 10% SDS, 50% glycerol, 25% 2-mercaptoethanol, 0.02% bromophenol). Then protein samples were loaded on 8% SDS-PAGE gel followed by standard Western blotting (Trans-blot semi dry transfer cell machine: Biorad 470-3940, Immobilon-PSQ transfer membrane: Millipore). Blots were incubated with Nav1.5 (1:500, Sigma-Aldrich, S0819) or Calnexin (1:2000, Millipore, VWR208880) overnight at 4 °C. A HRP-conjugated secondary antibody (1:2500, GE Healthcare Life Sciences) was applied and incubated for 2 h at room temperature. Immuno-reactive bands was visualized using chemo-luminescent ECL-PLUS Western blot reagents (Amersham Pharmacia Biotech) and luminescence recording was performed with the ImageQuant LAS 4000 Bio-molecular image analyzer. Protein concentration was quantified using ImageJ. Source data of uncropped blots are provided as a Source Data file.

**Sanger sequencing**. DNA sequencing was performed with BigDye Terminator Cycle Sequencing Kit (Applied Biosystems). The BigDye Cycle Sequence

sequencing reaction utilized the following cycling parameters: initial denaturation at 96 °C (10 s), denaturation at 96 °C (30 s), with an intervening 40 cycles of prime annealing at 50 °C (15 s), elongation 60 °C (4 min). Sequencing is outsourced to the AMC Sequence Facility. Results are retrieved via LIMS from the AMC Sequence Facility and analyzed with CodonCode Aligner (CodonCode Corporation).

**Mouse transgenic enhancer assay**. The regulatory region RE6-9 (chr9: 119377827-119389749) was PCR amplified from BAC RP23-198L19 using the primers (forward—GTTACCCAACAAGTCAGAGAAACAG and reverse—TCTGCAACCCTATAGGTGGAACA) and was cloned into the TOPO vector (Invitrogen). Insert RE6-9 was validated by restriction enzyme digestion analyses and sequencing, and recombined into a Hsp68-LacZ-Gateway reporter vector using Gateway LR Clonase II Enzyme Mix kit (Invitrogen)[69]. Generation of transgenic embryo staining was performed in accordance with protocols approved by Cyagen Biosciences Inc. (Santa Clara, CA).

**In vivo ECG recording**. Animals were anaesthetized with 1.5% isoflurane. Electrodes were placed at the right (R) and left (L) armpit and the left groin (F). A reference electrode was placed at the right groin. ECGs were recorded (Biosemi, Amsterdam, the Netherlands; sampling rate 2048 Hz, filtering DC 400 kHz (3 dB)) for a period of 2 min. From these leads, a standard three-lead ECG was calculated as follows: I = L − R, II = F − R, III = F − L, aVR = R − (L + F)/2, aVL = L − (R + F)/2, and aVF = F − (L + R)/2[70].

**Langendorff experiments**. Mice were stunned by inhalation of CO₂ and killed by cervical dislocation, after which the heart was excised, cannulated, mounted on a Langendorff perfusion setup, and perfused at 37 °C with Tyrode's solution (128 mmol/L NaCl, 4.7 mmol/L KCl, 1.45 mmol/L CaCl₂, 0.6 mmol/L MgCl₂, 27 mmol/L NaHCO₃, 0.4 mmol/L NaH₂PO₄, and 11 mmol/L glucose (pH maintained at 7.4 by equilibration with a mixture of 95% O₂ and 5% CO₂)). Ex vivo ECGs were recorded (Biosemi, Amsterdam, the Netherlands; sampling rate 2048 Hz, filtering DC 400 kHz (3 dB)) and analyzed using LabChart Pro (by Mitchells formula). Activation patterns were measured during sinus rhythm and ventricular and atrial pacing at a basic cycle length of 120 ms (twice the diastolic stimulation threshold). The effective refractory period of the AV node was determined by atrial pacing and reducing the coupling interval of a premature stimulus (after 8 stimuli at basic cycle length of 120 ms) in steps of 5 ms until activation of the ventricle failed[70].

**Luciferase assays**. RE9 (chr9: 119378700-119379271), major allele, T or minor allele, C, were cloned in pGL2-basic-luciferase construct that contains a minimal promoter. Luciferase constructs were co-transfected with pcDNA3.1 constructs expressing Gata4, Nkx2-5, Tbx20. Constructs were transfected in H10 cells with the PEI transfection reagent (Brunschwick). Standard transfections used 1.0 μg of reporter construct (or control reporter construct) co-transfected with 3 ng phRG-TK Renilla vector (Promega) as normalization control. Luciferase measurements were performed using a Promega Turner Biosystems Modules Multimode Reader Luminometer.

**Statistical analysis**. Results are expressed as mean ± SEM. Two group mean differences were evaluated using the unpaired Student's *t* test and more than two groups were compared using one-way ANOVA followed by post hoc Bonferroni test. For luciferase assays statistics was validated using a two-way ANOVA test. Statistical significance was considered for $P < 0.05$.

For 4C-seq data, we identified a set of 11 (119055000, 119195000, 119368500, 119398500, 119415000, 119543000, 119648000, 119661000, 119810000, 119833000, 119944000) and 10 (119055000, 119205000, 119368500, 119493000, 119648000, 119661000, 119810000, 119833000, 119944000) genomic locations at chromosome 9 from Scn5a promoter and Scn10a RE1 4C profiles respectively, to be tested for difference in mean in contact profiles. Anchors of 10 Kb were created at each position (~5 Kb around each of the genomic location as indicated), and the sequence depth normalized reads coverage were summed up from each restriction fragment in the anchor. An unpaired Student's *t* test was applied, using R 3.5.3.

**Reporting summary**. Further information on research design is available in the Nature Research Reporting Summary linked to this article.

## Data availability

RNA-seq data and 4C-seq data were deposited to the Gene Expression Omnibus under accession number GSE129067. All other relevant data supporting the key findings of this study are available within the article and its Supplementary Information files or from the corresponding author upon reasonable request. Detailed information on stats report of the 4C-seq data per replicate is available in Supplementary Table 5. Data underlying Figs. 4a, b, d–f; 5c–g; 6b, c, e, f, h, i; 7d–g and Supplementary Figs. 8a, b; 9a–d; 10c in this study are provided as a Source Data file. A reporting summary for this Article is available as a Supplementary Information file.

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

## Acknowledgements

This study was supported by CVON HUSTCARE, Charles River grant and by a Leducq Foundation grant (14CVD01).

## Author contributions

J.C.K.M. performed the 4C experiment, RNA-sequencing, in vivo mouse experiments, analyzed and interpreted the data, and wrote the manuscript. R.A.M. performed the electrophysiological experiments, analyzed and interpreted the data. M.B. performed the in vitro H10 cells experiment and in vivo mouse experiments, analyzed and interpreted the data. C.R.E.H. performed the 4C experiment, analyzed and interpreted the data. C.J. performed the electrophysiological experiments, analyzed and interpreted the data. V.W. performed the in vivo mouse experiments, analyzed and interpreted the data. V.B. analyzed and interpreted the 4C data. W.d.L. analyzed and interpreted the 4C data. P.B. contributed to data interpretation. B.J.B. performed the electrophysiological experiments, analyzed and interpreted the data. V.M.C. conceived the project, designed the experiments, interpreted the data and wrote the manuscript.

## Competing Interests

W.d.L. is founder and shareholder of Cergentis.
