## [Peer Review File · Nature Communications]

Reviewers' comments:

Reviewer #1 (Remarks to the Author):

The authors examine further the molecular components/structure of enhancer regions regulating SCN5A expression by generating several relatively large deletions in the mouse genome as well as the effects of these deletions on Scn5a expression, cardiac conduction and embryonic development.

The authors conclude that this "enhancer cluster in the SCN5A-SCN10A locus" "controls the conformation of the locus and Scn5a expression" suggesting that genetic variants in this region "may influence cardiac function".

Assessment

The authors build on previous studies investigating the properties of an enhancer cluster of regulatory elements (REs) in the SCN5A-SCN10A locus regulating Scn5a expression. Collectively, previous work from this group linked genetic variants in these non-coding enhancer regions to Scn5a-related electrical disturbances and arrhythmias. The stated purpose of this study is to "understand the relation between genetic variants found in non-coding regions in the SCN5A-SCN10A locus and the susceptibility to arrhythmias".

The studies demonstrate that this enhancer region modulates electrical function and influences heart development in association with changes in Scn5a expression using large deletions introduced selected REs. Overall, the general observation that RE deletions affect the electrophysiological and developmental properties of the heart seem rather predictable given previous publications from this group, which limits somewhat the novelty of the studies. Nevertheless, some of the results are difficult to reconcile with previous studies. For example, RE6 deletions had little to no effect on the Scn5a expression, electrophysiology or development, despite links between this region and electrical variability as well as arrhythmias (such as the rs6781009 variant). Additionally, the RE6-9 deletion had no effects on Scn5a expression in the atria. Unfortunately, neither of these unexpected findings, which are in some respects the most interesting observations, are discussed by the authors. While the results clearly show that RE9 can modulate Scn5a expression as well as the electrophysiology and development of the heart, the molecular underpinnings of these observations are only superficially investigated.

Other comments

1. The authors should include more details on the Hi-C data and analyses used to determine the TADs boundaries of the SCN5A-SCN10A locus.
2. Based on the results in Figure 3, the authors conclude that "RE1 and Wdr48 are brought in close proximity". The basis for this conclusion is unclear from the figure. Also, the authors should provide statistical evidence for the conclusions made from the studies summarized in Figure 3.
3. The evidence for altered electrical function in the ventricles of various mutants is weak. The authors should perform conduction maps in isolated hearts to determine the functional effects of reduced Scn5a expression.
4. No statistics are performed to conclude that AV block occurs in the mutant hearts. AV block is not an uncommon observation in isolated mouse hearts. Further analyses are needed. The authors might consider examining the effects of carbachol and beta-agonists on PR intervals.
5. As far as I can tell, EMERGE is a tool developed by the authors and not a standard analysis performed in the field. Is this program widely used? EMERGE plots appear repeatedly in the manuscript. The authors should describe in greater detail how these plots were generated for the different figure.

Reviewer #2 (Remarks to the Author):

In the submitted manuscript, Man et al. analyse the functional role of gene regulatory elements in the vicinity of the gene (SCN5A) encoding the major cardiac voltage-gated sodium channel. Genetic mapping in humans has identified a large number of variants in non-coding regions of the genome which are associated with cardiac arrhythmia. This study investigates the functional role of a particular region downstream of the SCN5A gene and identifies a novel regulatory element which modulates SCN5A expression and cardiac electrophysiology in mouse models in vivo. The study elegantly applies mouse genetics to generate targeted deletions of regulatory regions as well as state of the art 3-dimensional epigenetic analyses to determine chromatin structure.

Specific points:

1) Figure 1: In panel b, analysing promoter capture HiC data from iPS-derived cardiac myocytes, no interaction between the SCN5A gene promoter and the identified regulatory region downstream of SCN5A are visible. SCN5A promoter interactions seem to cross TAD boundaries into the next TAD. Please clarify.

2) Figure 1e: Please add size markers and indicate which embryonic stages are displayed and how many replicates were analysed. Please explain numbers on top of the hearts (4/5 etc.) in the legend. Similarly, add size markers and age info to the legend of Figure 2c.

3) Figure 3a, b: The changes in interactions between the SCN5A promoter and the RE6-9 locus associated with targeted deletion of the RE6-9 region are rather small. One reason may indeed be heterozygosity. However, analysing cardiac tissue (rather than cardiac myocytes) adds another important level of complexity. To further substantiate the findings, it would be important to validate the interaction between SCN5A promoter and RE6-9 in WT and mutant cardiac myocytes from mouse hearts. Furthermore, in Fig. 3a the interaction peak at the RE6-9 regions seems to be cut off, not allowing to see the full difference between WT and mutant signal. Please magnify this genomic region and provide quantification and statistical analyses for the detected interactions and differences between genotypes.

4) The manuscript concludes that Tbx20 (but not Tbx3) is an import factor controlling SCN5A expression via modulating activity of the RE9 regulatory element - based on experiments performed in H10 cells and displayed in Fig. 8c. Thus, it would be important to demonstrate that Tbx20 is also binding to the endogenous RE9 region in cardiac myocytes in vivo. Additional ChIP-seq data for Tbx20 would be essential to show binding.

5) Full details of the sequencing statistics for 4C-seq (Figure 3a, b) should be given in the manuscript and all sequencing data should be deposited in a public data base.

6) The predicted binding sites for the investigated transcription factors, especially for Tbx factors, should be added to the figures (or to a higher magnification figure together with ChIP-seq data for TFs).

Reviewer #3 (Remarks to the Author):

The manuscript by Man et al., studies the implication on an enhancer cluster on the regulation of SCN5A, a gene encoding the major cardiac sodium channel. Based on previous enhancer reporter assays, this regulatory unit appears to be composed of individual elements that also overlap with GWAS hits for variants associated to abnormal cardiac function. Through genome editing, the

authors perform a series of deletions to dissect this enhancer cluster in depth, performing downstream transcriptomic and chromatin conformation capture assays (4C-seq).

The study is interesting as it demonstrates the existence of a regulatory hierarchy within the individual elements of the enhancer cluster that is essential to control *Scn5a* expression. In addition, it provides a framework for the interpretation of pathogenic variants at the locus. However, the current manuscript raises several concerns regarding the methodology and data interpretation that need to be addressed by the authors.

Major comments:

1- 4C-seq experiments show that the interaction preferences within the *Scn5a* TAD are altered. However, the interaction change observed with RE6-9 and *Scn5a* in panel 3a is not quite prominent and might well be attributed to variation between replicates or data processing. How did the authors deal with the fact that 17kb (and approximately 60 4C-fragments) were deleted in heterozygosity in the mutants compared to wild type controls. Can the observed differences be attributed to the lack of signal in the deleted region? This aspect urges clarification.

2- The 4C profile from mutants seem to be overall slightly higher than the wild type controls. This is quite evident outside the TAD region and could explain the gained interaction with *Wdr48*. The authors claim that this is a biological effect but it might also well be a normalization problem. During the normalization process the frag-end with the highest coverage (presumably corresponding to the self-circularization PCR product) was excluded. But other fragments around the viewpoint also display a high number of reads (they are above the upper limit of the Y-axis). Was there also an incomplete digestion band in the 4C PCRs? Was this fragment removed from the analysis?

Since 4C-seq is a PCR-based method, overrepresented fragments around the viewpoint can easily create sample-to-sample amplification biases that can impact the normalization. Since the displayed differences are marginal on some interactions, the authors should demonstrate that these are not technical issues.

The experiments were performed in up to 3 different replicates for each condition. Their hypothesis could be supported by displaying all profiles in a supplementary figure to show that there is not such PCR bias in individual samples.

3- "These data indicate that heterozygous deletion of RE6-9 results in a monoallelic expression of *Scn5a*..."

There is solid evidence within the text that this is definitely not the case. Homozygous embryos still express around 30% of *Scn5a* compared to wild type controls (Fig 7d and f). Categorizing such levels as low is questionable in this case and the experiment demonstrates that gene copies in cis with the mutation can effectively fire transcription.

Another evidence against monoallelic expression is the absence of changes on the QRS interval in heterozygous RE6-9 mutants, which contrasts with the increase observed in heterozygous *Scn5a* mutants. Such effect would be expected if the expression of the allele containing the regulatory mutation was entirely suppressed (and no other gene was misregulated as suggested by the authors on fig 4d-f).

4- "Enhancer cluster RE6-9 is required for the intra TAD three-dimensional organization"

An obvious question arising from this section is if RE6-9 is bound by architectural factors such as CTCF or cohesin that could explain the effects in 3D spatial organization upon deletion.

The authors should include ChIP-Seq tracks for such factors in the corresponding figure. They can be found in ENCODE datasets (ENCODE., Nature 2012). Also, comment this aspect in results and discussion.

5- The developmental stage where all experiments were performed is not indicated.

This aspect is especially problematic for the RNA-seq experiment of ventricles. It is quite

remarkable that only Scn5a shows a clear misregulation on the analysis and not a single downstream gene is prominently affected.

At which time point was the experiment performed? When does Scn5a starts to be expressed in developing hearts? In other words, could be that the stage chosen is in close temporal proximity to when Scn5a starts to be expressed during normal development, so that only expression differences on this gene are detected?

6- Cited references are often not adequate.

Prominent examples are found in the introduction, where the TAD concept is presented. None of the studies where TADs were first reported are cited (Dixon et al., Nature 2012; Nora et al., Nature 2012).

In the following sentence, the citation of Montavon et al., 2012 does not really fit with the information mentioned. Furthermore, when chromosome conformation capture technologies are cited, landmark papers are not referenced.

These are just a few examples and the authors should make a thorough revision effort to provide accurate references across the entire text.

7- "... the resolution of such maps is still low..."

4C-seq and Promoter Capture Hi-C can easily generate high-quality maps at fragment resolution. Also, Bonev et al., Cell 2017, provided Hi-C maps at 750bp resolution, which can be considered as very high and sufficient to identify regulatory interactions.

8- "...many interactions are not seen..."

Could the authors explain the meaning of this statement and provide appropriate references to the claim?

9- "TADs are stable between different cell types and highly conserved between species"

Most TADs can be considered as stable among cells and species but this is not a universal rule. Around 30% of TADs show differences between cell types (Dixon et al., Nature 2012). Same for species.

10- "These findings suggest that the enhancer cluster RE6-9 is partially redundant during development".

Partially redundant with which other elements? Partially referred to spatial or temporal pattern? Please state the interpretation more clear.

11- "These findings suggest that RE6-8 is functionally redundant during development."

Same as above. Clarification is also required here.

12- "Genetic variants affecting the activity of the enhancer cluster"

Conclusions and interpretation are missing in this section.

13- "REs such as enhancers are usually found to act in a modular and additive fashion in that the each RE of a multi-component RE system adds its spatio-temporal activity pattern to that of the target gene(s)"

Many studies have shown that enhancers also act in different modes than just modular and additive. Authors should acknowledge these mechanisms and provide adequate references. This section of the discussion needs to be improved to frame the results in a better context with the current knowledge on the field.

14- "Recently, it was shown that such REs in the mouse genome act redundantly in vivo; individual removal of an RE from the genome causes only slightly reduced expression of the target gene that does not result in phenotypic changes"

Again, there are various example that show different modes of enhancer activity. See Gonen et al., Science 2018 or Shin et al., Nature Genetics 2016 (this last one is already cited by the authors).

They are both examples of hierarchy within an enhancer cluster, like the case here presented.

Minor comments:

1- Figure 2c would benefit from including in situ hybridizations for Scn5a and Scn10a to illustrate the similarities with the results of the enhancer reporter assays. In addition, the total number of embryos displaying positive staining and analyzed for RE6-9 should be indicated.

2- Please indicate in which specific reference can be located each of the transgenic reported displayed in Figure 1e.

According to the cited references reporters were also performed for mouse RE9. It would be nice to include pictures side by side the human version of the tested enhancer, to appreciate the difference of activity between species, as well as to directly compare with the activity of RE6-9. Were also mouse enhancers tested for RE6 and RE7? If yes, please include it in the figure.

3- "We found that the Scn5a promoter contacts multiple sites within the TAD including the promoters of Scn10a and Exog, RE6-9, RE5 within Scn5a, and RE1 within Scn10a"
It would be good for the readers to give a brief introduction of what is known about RE5 and RE1.

4- Figure 5a is referenced out of place in the text.

5- "Expression" is misspelled in Figs 7f and g

Reviewer #4 (Remarks to the Author):

This manuscript investigates a region of DNA lying 3' to the SCN5A gene in both mouse and human. Through bioinformatic integration of a variety of resources, evidence is presented that a ~12Kb region has characteristics of a "super-enhancer". The enhancer is shown to be functional, and expressed in the mouse heart during development. Heterozygous deletion of the region ablates expression of the SCN5A allele that is in cis, without a large effect on other genes in the region. Heterozygous mice show differences in the ECG in vivo and in Langendorff perfusion. Homozygous deletion of the region is shown to be embryonic lethal, exhibiting a phenotype consistent with Scn5a loss of function. Sub-deletions give an indication that one part of the region ("RE9") has a greater input to gene regulation than other parts, with partial compensation during development. Alleles of a QRS associated human genetic variant, rs6810361, in the "RE9" region, are differentially affected by key cardiac transcription factors in reporter assays.

The manuscript contains a lot of careful work and it presents a strong case for the importance of this region in SCN5A gene regulation. The work will be of interest both to those studying genetics of rare cardiac arrhythmic disease, particularly Brugada syndrome, and to those studying the impact of common variants on cardiac electrophysiology. But, I think the authors have overstated their case in recommending genetic testing of variants in this region for Brugada syndrome as a result of this work – my interpretation is that their work rather highlights the extreme difficulty of extrapolating in this way.

I have some points for the authors' consideration.

1. Figure 1 could be more clearly presented. Multiple panels with some redundancy at present. The resolution is rather low for some panels also. Please consider highlighting the chromatin interactions that are of particular interest. Does the triangle plot, as I assume, represent LD in the region, or something else? Not labelled

2. Please make clear where the data underpinning Figure 1 originate. What for example is the input to the EMERGE predictions? It seems that many interactions are crossing the TAD domain,

are the boundaries correctly represented? The scales on Figure 1 are not clear. It's stated that PR-interval variants are more found in SCN10A region and QRS variants more distributed, but this isn't easy to see, the statement needs statistical support. Is there H3k27Ac data available for hESC-CM's that could be integrated?

3. Figure 1 is focused on human data but there is also a mouse heart enhancer reporter assay. For clarity that would fit better in figure 2 which is mouse data.

4. In figure 2, there is substantial difference between the three transgenic enhancer examples. The lowest is quite different from the upper two. How is this to be interpreted? Please explain if the construct will only express LacZ if all the enhancer regions are active, or only one? Given the findings when the region is dissected, this is important to know.

5. The RE6-9 deletion was constructed so that the 3' ends of EXOG and SCN5A were not disrupted. I did not see confirmation that this was successfully achieved and those genes were left intact.

6. Figures 3 a and b are interpreted to show that the RE6-9 hemizygotes exhibit a lesser degree of chromatin interaction from two viewpoints. The differences appear quite small, and there is no statistical test performed to support that the interactions are indeed different at the particular regions of interest between wildtype and hemizygote. I presume the greenish colour in the diagram indicates overlap but this is not stated.

7. The allele specific expression experiment is presented in figure 5b. This must, I think, represent sequence of expressed RNA, but it's not stated. It's something of a concern that the wildtype FVB alleles at Scn5a appear to have lower expression than the wildtype SPRET alleles, in the top panel, though it is convincing that they are undetectable in the FVB KO, which isn't the case for the other genes. Please clarify and address.

8. How do the authors interpret the lack of effect of RE6-9 deletion in the right atrium (Figure 5c)?

9. In the in vivo experiments, the difference between the PR intervals in the wildtype and hemizygous animals in the pseudo-ECG Langendorff experiment appears to be about 30%, but in vivo it is only 3%, a perhaps surprisingly small difference. Please place these differences in context.

10. Rs6810361 is a QRS duration associated variant, is it also a Brugada Syndrome associated variant? Why did the authors test a QRS duration variant when their mouse model had shown no effect on QRS duration but rather PR interval?

Reviewer #1

We thank the reviewer for the careful review and constructive comments. Please find below a detailed response to all points.

Comments

“Nevertheless, some of the results are difficult to reconcile with previous studies. For example, RE6 deletions had little to no effect on the *Scn5a* expression, electrophysiology or development, despite links between this region and electrical variability as well as arrhythmias (such as the rs6781009 variant). Additionally, the RE6-9 deletion had no effects on *Scn5a* expression in the atria. Unfortunately, neither of these unexpected findings, which are in some respects the most interesting observations, are discussed by the authors.”

Indeed, the very mild response of deletion of the RE6 component of the enhancer cluster was surprising, also in light of our previous observation that RE6 is a bona fide enhancer containing a presumably functional variant associated with conduction traits. We have discussed this issue in the revised Discussion section, page 13. In addition, we discuss the lack of response in the right atrial tissue in the context of RE redundancy in the revised Discussion, page 13.

Other comments

1. The authors should include more details on the Hi-C data and analyses used to determine the TADs boundaries of the *SCN5A-SCN10A* locus.

In revised Fig. 1a, we have presented a Hi-C map of the lymphoblastoid cell line GM12878 (Rao *et al.*, Cell 2014) for the *SCN5A-SCN10A* locus, which was adapted from the visualization tool HiGlass (Kerpedjiev *et al.*, Genome Biology 2018) and marked the TAD boundaries according to the interaction heatmap triangles. Furthermore, we used publically available Hi-C data and promoter capture Hi-C data from human iPSC-derived cardiomyocytes (Montefiori *et al.*, eLife 2018) to confirm the TAD domains of the *SCN5A-SCN10A* locus. These data are publically available at 3D Genome Browser and WashU EpiGenome Browser (see links below):

- <http://promoter.bx.psu.edu/hi-c/view.php>
- <http://epigenomegateway.wustl.edu/browser/?genome= hg19&publichub=Lindsey>

We also added a track of the cardiac CTCF occupancy (ENCODE Project Consortium. Nature. 2012; Davis *et al.*, Nucleic Acids Res. 2018), which often demarcates, amongst others, TAD boundaries (e.g. Ong CT and Corces, Nat Rev Genet. 2014; Nora *et al.*, Cell. 2017).

2. Based on the results in Figure 3, the authors conclude that “RE1 and Wdr48 are brought in close proximity”. The basis for this conclusion is unclear from the figure. Also, the authors should provide statistical evidence for the conclusions made from the studies summarized in Figure 3.

To address this comment, we have tested for difference in mean signal at 11 positions relative to the *Scn5a* promoter viewpoint and 10 positions relative to the RE1 viewpoint between wildtype (n=3) and RE6-9^{+/-} mutants (n=3). Furthermore, we have applied an unpaired student *t*-test using R 3.5.3 to each location. These results can be found in Supplementary Figure 4 for the *Scn5a* promoter viewpoint and Supplementary Figure 5 for the RE1 viewpoint. This analysis shows that intra-domain contacts between *Scn5a* promoter, *Scn10a* promoter, RE6-9, RE1 and RE5 are significantly reduced. In contrast, interactions outside the *Scn5a* TAD were not significantly different (*e.g.* location 1 and 2). We have addressed these findings in the Results section and removed the conclusion regarding increased contacts between the viewpoints and *Wdr48*.

3. The evidence for altered electrical function in the ventricles of various mutants is weak. The authors should perform conduction maps in isolated hearts to determine the functional effects of reduced *Scn5a* expression.

As requested by the reviewer we have performed additional experiments, recorded ECGs from additional mice, and measured ventricular conduction velocity using optical mapping. The results are shown in panel b, c and g-i, respectively, of the revised version of Fig. 6. Both PR interval and QRS duration were increased in both mutants. Longitudinal conduction velocity was lower in the left ventricle of RE6-9^{+/-} mice compared to wildtypes. The latter supports the observed QRS prolongation in *in vivo* ECGs taken from RE6-9^{+/-} mice.

4. No statistics are performed to conclude that AV block occurs in the mutant hearts. AV block is not an uncommon observation in isolated mouse hearts. Further analyses are needed. The authors might consider examining the effects of carbachol and beta-agonists on PR intervals.

To address this issue, we measured additional hearts. From in total 9 mice we recorded pseudo-ECGs from isolated Langendorff-perfused hearts (3 wildtype and 6 RE6-9^{+/-}). An unpaired student *t*-test showed PR interval to be significantly prolonged in the RE6-9^{+/-} mice (revised Fig. 6e). Moreover, RE6-9^{+/-} mice showed first-degree AV block (n=3) and second-degree type I AV block (Wenckebach, 4:3 and 3:2; n=3) whereas the wildtypes did not (revised Fig. 6d). In our hands AV block is highly uncommon in Langendorff-perfused mouse hearts (Boukens *et al.*, *Cardiovasc Res.* 2013, Aanhaanen *et al.*, *J Clin Invest.* 2011, Gillers *et al.*, *Circ Res.* 2015).

5. As far as I can tell, EMERGE is a tool developed by the authors and not a standard analysis performed in the field. Is this program widely used? EMERGE plots appear repeatedly in the manuscript. The authors should describe in greater detail how these plots were generated for the different figure.

We have published EMERGE in *Nucleic Acids Research* in 2016 (van Duijvenboden *et al.*, *Nucleic Acids Res.* 2016). The EMERGE tool is available for public use and it merges datasets considered informative, including

ATAC-seq, CHIP-seq, and homology, and uses a logistic regression framework, based on validated functional elements, to assign optimal weights to these datasets. The merged and weight signal can be peak called, the peaks predicting functional RE (van Duijvenboden *et al.*, Nucleic Acids Res. 2016). We have added Supplementary Figure 1 and Table 1 showing the EMERGE pipeline and the data sets used.

Reviewer #2:

We thank the reviewer for the positive assessment and the insightful comments. Please find below a detailed response to all points.

Specific points:

1) Figure 1: In panel b, analysing promoter capture HiC data from iPS-derived cardiac myocytes, no interaction between the SCN5A gene promoter and the identified regulatory region downstream of SCN5A are visible. SCN5A promoter interactions seem to cross TAD boundaries into the next TAD. Please clarify.

Promoter capture Hi-C identifies long-range interactions between promoters and non-coding regions with high resolution. However, Montefiori *et al.*, eLife 2018 mentions the technical and biological limitations of this technique and consequently interpreting the iPS-derived cardiomyocytes interaction dataset. It is likely that the promoter capture Hi-C dataset has not mapped all the interactions due to distance-dependent effect and sequencing depth. The regulatory region (RE6-9) lies approximately 100 kb downstream of the SCN5A promoter and might not have been captured using the SCN5A promoter as viewpoint. Nevertheless, we have confirmed that SCN5A/Scn5a promoter interacts with RE6-9 analysing 4C-seq data from human ventricular tissue and mouse adult heart (van der Harst *et al.*, J Am Coll Cardiol. 2016; van den Boogaard *et al.*, JCI 2014). In addition, a virtual 4C-seq tool which uses an algorithm to integrate human atrial and ventricular Hi-C datasets to generate computational interaction data for a viewpoint of interest (Bianchi, de Laat, manuscript in preparation), has identified an interaction between this regulatory element and the SCN5A promoter (Reviewer Figure 1). Indeed, interaction between the Scn5a promoter and sequences in the adjacent TAD are observed. These may be real, as the TAD boundaries are not strict. However, the interaction may not be functional, as enhancer deletion from the mouse genome does not affect gene expression outside the TAD. We have added this information to the revised Discussion section.

Reviewer Figure 1. Virtual 4C-seq plot of human LA and LV, bait set on SCN5A promoter (Valerio Bianchi and Wouter de Laat; unpublished data)

2) Figure 1e: Please add size markers and indicate which embryonic stages are displayed and how many replicates were analysed. Please explain numbers on top of the hearts (4/5 etc.) in the legend. Similarly, add size markers and age info to the legend of Figure 2c.

In revised Fig. 1c, mouse embryos for Hsp68/LacZ enhancer reporter assay of human RE6, RE7 and RE9 were isolated at stage E10.5. Numbers at the bottom of the panels indicate the number of LacZ expressing hearts of total number of transgenic embryos for each construct. In revised Fig. 2c, the activity of mouse RE6-9 was analysed in E11.5 embryos. In revised Fig. 1c, Fig. 2c and Fig. 7a-c we have added size markers.

3) Figure 3a, b: The changes in interactions between the SCN5A promoter and the RE6-9 locus associated with targeted deletion of the RE6-9 region are rather small. One reason may indeed be heterozygosity. However, analysing cardiac tissue (rather than cardiac myocytes) adds another important level of complexity. To further substantiate the findings, it would be important to validate the interaction between SCN5A promoter and RE6-9 in WT and mutant cardiac myocytes from mouse hearts. Furthermore, in Fig. 3a the interaction peak at the RE6-9 regions seems to be cut off, not allowing to see the full difference between WT and mutant signal. Please magnify this genomic region and provide quantification and statistical analyses for the detected interactions and differences between genotypes.

We agree with the reviewer that the changes in interaction signal are limited because of heterozygosity. In the 4C-seq experiment, we used whole hearts of RE6-9^{+/-} mice, including different cell types (e.g. cardiomyocytes, fibroblasts, endocardial cells), which might indeed further diluted cardiomyocyte-specific interactions. Despite these limitations, statistical analysis on the 4C-seq data showed that the interactions between *Scn5a* promoter, *Scn10a* promoter, RE6-9, RE1 and RE5 are significantly reduced in RE6-9^{+/-} mice compared to wildtype mice. Please see revised Fig. 3, Supplementary Figure 3 and Supplementary Figure 4 for statistical analysis of the interactions.

Isolation and purification of cardiomyocytes from adult hearts suitable for 4C-seq was not possible. We have attempted a quantitative 3C analysis on PCM-1+ cardiomyocyte nuclei isolated and FACS purified from adult heart, but this experiment was also not successful. Nevertheless, we hope the reviewer agrees that the current analysis of the 4C-seq data provides sufficient evidence for reduced interactions between REs and promoters distal from RE6-9. And we have removed the claim that conformation is specifically altered in cardiomyocytes, or heart, as we have not measured this.

4) The manuscript concludes that Tbx20 (but not Tbx3) is an import factor controlling SCN5A expression via modulating activity of the RE9 regulatory element - based on experiments performed in H10 cells and displayed in Fig. 8c. Thus, it would be important to demonstrate that Tbx20 is also binding to the endogenous RE9 region in cardiac myocytes in vivo. Additional CHIP-seq data for Tbx20 would be essential to show binding.

To address this comment, we have included Tbx20 ChIP-seq data from E11.5 mouse hearts (Boogerd *et al.*, J Clin Invest. 2018) in revised Fig. 2b and revised Supplementary Figure 10 (see also Reviewer Figure 2). In this data set we identified a Tbx20 occupied site in RE9 at the predicted location overlapping rs6810361.

Reviewer Figure 2. *Tbx20* ChIP-seq from E11.5 heart, red line RE9

5) Full details of the sequencing statistics for 4C-seq (Figure 3a, b) should be given in the manuscript and all sequencing data should be deposited in a public data base.

We have deposited the RNA-seq data and 4C-seq data to the Gene Expression Omnibus under the accession number GSE123440 and GSE129067. These accession numbers are included in the Methods, section *Data availability* of the revised manuscript. Furthermore, we have included Supplementary Table 2 showing the Stats report of the 4C-seq data per replicate.

6) The predicted binding sites for the investigated transcription factors, especially for Tbx factors, should be added to the figures (or to a higher magnification figure together with ChIP-seq data for TFs).

We have added revised Fig. 2d which shows a zoom in on predicted transcription factor binding sites for Tbx5, Tbx20 and Gata4 in RE9 including sequence conservation between species and JASPAR consensus sites.

Reviewer #3:

We thank the reviewer for positive assessment of our study and for providing constructive comments that have helped us to improve the manuscript. Below, a point-by-point response to each of the comments is given.

Major comments:

1- 4C-seq-seq experiments show that the interaction preferences within the *Scn5a* TAD are altered. However, the interaction change observed with RE6-9 and *Scn5a* in panel 3a is not quite prominent and might well be attributed to variation between replicates or data processing.

How did the authors deal with the fact that 17kb (and approximately 60 4C-seq-fragments) were deleted in heterozygosity in the mutants compared to wild type controls. Can the observed differences be attributed to the lack of signal in the deleted region? This aspect urges clarification.

To address the first comment, we have included Supplementary Figure 6 (*Scn5a* promoter viewpoint) and Supplementary Figure 7 (RE1 viewpoint) showing the overlay for each combination per replicate (blue signal = WT; orange signal = RE6-9^{+/-}; grey signal = shared). Independent of the combination of a single replica, we observed that the normalized 4C-seq signal is always increased or reduced in one condition (WT) versus the other (RE6-9^{+/-}), which demonstrate the reproducibility of the 4C-seq experiment. Furthermore, we have performed a statistical analysis (unpaired student *t*-test using R 3.5.3) to determine whether the contact profiles of 11 positions relative to the *Scn5a* promoter viewpoint and of 10 positions relative to the RE1 viewpoint were significantly different between wildtype (n=3) and RE6-9^{+/-} mutants (n=3). These results can be found in Supplementary Figure 3 for the *Scn5a* promoter viewpoint and Supplementary Figure 4 for the RE1 viewpoint. This analysis shows that interactions between *Scn5a* promoter, *Scn10a* promoter, RE6-9, RE1 and RE5 are significantly reduced in mutants. In contrast, interactions outside the *Scn5a* TAD were not significantly different (*e.g.* location 1 and 2). We have addressed these findings in the Results section, page 5-6, line 147-161.

In the second comment the reviewer rightfully points out the heterozygous deletion of RE6-9 may contribute to the weaker signal at that location in RE6-9^{+/-} mouse hearts. Nevertheless, the viewpoints and signals at RE1, RE5, *Scn10a* promoter *etc.*, where we see reduction of interactions in the RE6-9^{+/-} mouse hearts, are not affected by the deletion. We have addressed this point in the Results section, page 5-6, line 147-161.

2- The 4C-seq profile from mutants seem to be overall slightly higher than the wild type controls. This is quite evident outside the TAD region and could explain the gained interaction with *Wdr48*. The authors claim that this is a biological effect but it might also well be a normalization problem. During the normalization process the frag-end with the highest coverage (presumably corresponding to the self-circularization PCR product) was excluded. But other fragments around the viewpoint also display a high number of reads (they are above the upper limit of the Y-axis). Was there also an incomplete digestion band in the 4C-seq PCRs? Was this fragment removed from the analysis?

There was no incomplete digestion band in the 4C-seq PCRs. Therefore, we have reanalyzed the 4C-seq data and recreated new overlays based on the signal at the level of a single restriction fragment (revised Fig. 3a and b). In these 4C-seq plots, blue signal represents WT, orange signal represents RE6-9^{+/-}, and gray signal shows the overlapped signal for both genotypes. We observe an equal baseline signal distribution for both wildtype and mutants outside the TAD region, except at positions where chromatin interaction is reduced or gained.

Since 4C-seq-seq is a PCR-based method, overrepresented fragments around the viewpoint can easily create sample-to-sample amplification biases that can impact the normalization. Since the displayed differences are marginal on some interactions, the authors should demonstrate that these are not technical issues.

The experiments were performed in up to 3 different replicates for each condition. Their hypothesis could be supported by displaying all profiles in a supplementary figure to show that there is not such PCR bias in individual samples.

Thank you for this suggestion. We have included Supplementary Figure 5 (*Scn5a* promoter viewpoint) and Supplementary Figure 6 (RE1 viewpoint) showing the overlay for each combination per replicate (blue signal = WT; orange signal = RE6-9^{+/-}; grey signal = shared). Independent of the combination of a single replica, we observed that the normalized 4C-seq signal is always increased or reduced in one condition (WT) versus the other (RE6-9^{+/-}), which demonstrates the reproducibility of the 4C-seq experiment. Furthermore, we have applied an unpaired student *t*-test to each location using R 3.5.3. These results can be found in Supplementary Figure 3 for the *Scn5a* promoter viewpoint and Supplementary Figure 4 for the RE1 viewpoint. The analysis shows that contacts between *Scn5a* promoter, *Scn10a* promoter, RE6-9, RE1 and RE5 are significantly reduced. In contrast, interactions outside the *Scn5a* TAD were not significantly different (*e.g.* location 1 and 2). We have addressed these findings in the Results section, page 5-6, line 147-161.

3- “These data indicate that heterozygous deletion of RE6-9 results in a monoallelic expression of *Scn5a*...” There is solid evidence within the text that this is definitely not the case. Homozygous embryos still express around 30% of *Scn5a* compared to wild type controls (Fig 7d and f). Categorizing such levels as low is questionable in this case and the experiment demonstrates that gene copies in cis with the mutation can effectively fire transcription.

Another evidence against monoallelic expression is the absence of changes on the QRS interval in heterozygous RE6-9 mutants, which contrasts with the increase observed in heterozygous *Scn5a* mutants. Such effect would be expected if the expression of the allele containing the regulatory mutation was entirely suppressed (and no other gene was misregulated as suggested by the authors on fig 4d-f).

Our study suggests that the regulatory function of RE6-9 is partially redundant during development as E10.5 embryos homozygous for the deletion indeed still express approximately 30% of wildtype *Scn5a* levels. In the postnatal heterozygous deletion carriers, however, the expression levels in the hearts (except for the right atrium) is reduced to 50%, suggesting expression from the allele with the deletion is strongly

reduced (close to zero). To address this issue further, we have included an allele-specific expression experiment (SPRET;FVB^{ΔRE6-9} F1 hybrid mice; FVB is the background we use for genome editing, SPRET mice have unique SNPs in the *Scn5a* locus suitable for allele specific detection of mRNA). In this experiment, the FVB *Scn5a* transcript is undetectable in SPRET;FVB^{ΔRE6-9} F1 hybrid mice, whereas other genes in the TAD were not affected (Fig. 5b). Furthermore, additional *in vivo* ECG experiments demonstrate prolongation of PR interval and QRS duration from RE6-9^{+/-} mice (revised Fig. 6b, c). The latter supports the observed decrease in longitudinal conduction velocity in the left ventricle of RE6-9^{+/-} mice (revised Fig. 6g, h). Together, these findings indicate reduced or monoallelic expression of *Scn5a* in RE6-9^{+/-} mice similar to heterozygous *Scn5a* mutants (Papadatos *et al.*, PNAS 2002). We have addressed these points in the revised Results section for better clarification.

As an additional point, recently, the process of transcriptional adaptation (loss of function mutant genes induce the expression of genes with sequence homology, such as paralogous genes) as one of the mechanisms underlying genetic robustness was discovered (Rossi *et al.*, Nature. 2015; El-Brolosy *et al.*, Nature. 2019). Transcription and mRNA decay of the mutated gene was found to be required. Therefore, we hypothesized that the 7-bp deletion in *Scn5a* exon 3 (*Scn5a*^{+Δ7}), which shows mRNA decay, would cause upregulation of other genes of the voltage-gated sodium channel alpha subunit family, whereas deletion of RE6-9, which causes transcriptional reduction, would not. We performed quantitative RT-PCR on 8-week old ventricles of *Scn5a*^{+Δ7} mice, and observed an upregulation of *Scn1a* and *Scn3a*, respectively. Furthermore, there was limited or no increase in *Scn2a* and *Scn10a* in *Scn5a*^{+Δ7} mice. Interestingly, *Scn4a* expression was decreased (Supplementary Figure 8). However, in 8-week-old ventricles of RE6-9^{+/-} mice we did not observe a change in expression of these genes (except for *Scn5a* and *Scn10a*) compared to wildtype (Supplementary Figure 8). Together, these data suggest that loss of function caused by an *Scn5a* coding mutation triggers transcriptional adaptation, whereas loss of function caused by an enhancer deletion does not.

4- “Enhancer cluster RE6-9 is required for the intra TAD three-dimensional organization”

An obvious question arising from this section is if RE6-9 is bound by architectural factors such as CTCF or cohesin that could explain the effects in 3D spatial organization upon deletion.

The authors should include CHIP-Seq tracks for such factors in the corresponding figure. They can be found in ENCODE datasets (ENCODE., Nature 2012). Also, comment this aspect in results and discussion.

We have added human and mouse CTCF CHIP-seq track (ENCODE Project Consortium. Nature. 2012; Davis *et al.*, Nucleic Acids Res. 2018) in revised Fig. 1a and Fig. 2a, b, indicating RE6-9 is not occupied by CTCF in the heart. Furthermore, we have commented on the role of architectural factors such as CTCF and cohesin in the three-dimensional organization of the *Scn5a-Scn10a* locus in RE6-9^{+/-} mice in the Results (page 5, line 132-136, and Discussion sections (page 12, line 351-356).

5- The developmental stage where all experiments were performed is not indicated.

This aspect is especially problematic for the RNA-seq experiment of ventricles. It is quite remarkable that only *Scn5a* shows a clear misregulation on the analysis and not a single downstream gene is prominently affected.

At which time point was the experiment performed? When does *Scn5a* starts to be expressed in developing hearts? In other words, could be that the stage chosen is in close temporal proximity to when *Scn5a* starts to be expressed during normal development, so that only expression differences on this gene are detected?

This is an interesting point we did not sufficiently clarify or discuss in the manuscript. The expression profile of *Scn5a* during heart development has been established (Remme *et al.*, Basic Res Cardiol. 2009). *Scn5a* expression starts in the embryonic heart at stage E9.5, increases towards stage E11.5 and further increases gradually after birth (Domínguez *et al.*, Cardiovasc Res. 2008). The distribution pattern of *Scn5a* mRNA expression and Nav1.5 protein expression is similar in adult and embryonic (E14.5) mouse hearts (Remme *et al.*, Basic Res Cardiol. 2009). It is highly expressed in the (prospective) atrioventricular bundle and trabecular ventricular myocardium and in the atrial body, and almost absent from the sinus node, atrioventricular canal/node. Homozygous *Scn5a* mutants die at E10.5, showing the requirement of *Scn5a* during embryogenesis (Papadatos *et al.*, PNAS 2002). The qPCR data in revised Fig. 7d, f is derived from E10.5 RE6-9^{-/-} mutants and wildtype controls. The RNA-seq analysis in Fig. 4 was performed on ventricular samples of RE6-9^{+/-} mice and wildtype controls 14 days after birth. The qPCR analysis shown in revised Fig. 5 were performed on adult (8-10 weeks after birth) RE6-9^{+/-} mice and wildtype controls. Taken together, the quantification of expression was performed well after expression of *Scn5a* is initiated and is necessary for development and heart function. We have indicated the developmental stages for the various transgenic mouse models used in the revised Figures and Results section. Furthermore, in the Discussion section we discuss the observation that prolonged reduction or loss of *Scn5a* expression, a gene essential for cardiac conduction and associated with arrhythmia syndromes such as Brugada in humans, does not affect expression of gene programs secondary to phenotypic changes such as conduction slowing (see revised Fig. 6; Discussion page 13).

6- Cited references are often not adequate.

Prominent examples are found in the introduction, where the TAD concept is presented. None of the studies where TADs were first reported are cited (Dixon *et al.*, Nature 2012; Nora *et al.*, Nature 2012).

In the following sentence, the citation of Montavon *et al.*, 2012 does not really fit with the information mentioned. Furthermore, when chromosome conformation capture technologies are cited, landmark papers are not referenced.

These are just a few examples and the authors should make a thorough revision effort to provide accurate references across the entire text.

We apologize for this oversight, in some cases we have cited more recent papers in which these and other papers are cited, or reviews. To address this issue, we have thoroughly revised the manuscript and provided accurate references to original work, including the ones mentioned by the reviewer. Please note the total number of references we are allowed to include is 70. Therefore, in some cases we cite reviews.

7- "... the resolution of such maps is still low..."

4C-seq-seq and Promoter Capture Hi-C can easily generate high-quality maps at fragment resolution. Also, Bonev et al., Cell 2017, provided Hi-C maps at 750bp resolution, which can be considered as very high and sufficient to identify regulatory interactions.

To address this, we have included reference (Bonev *et al.*, Cell 2017) and adjusted the sentence. Furthermore, we have revised Fig. 1a by presenting a high resolution Hi-C map of the lymphoblastoid cell line GM12878 (Rao et al., Cell 2014) for the *SCN5A-SCN10A* locus which was adapted from the visualization tool HiGlass (Kerpedjiev et al., Genome Biology 2018). In the mouse track (revised Fig 2) we have included a bar showing the TAD as indicated by the data from Bonev *et al.*, Cell 2017.

8- "...many interactions are not seen..."

Could the authors explain the meaning of this statement and provide appropriate references to the claim?

Given the huge number of possible interacting fragments, sparsity of fragment pairs detected even at high sequencing depths, requirement of binning, noise, the resolution of most Hi-C data sets is limited and even PChi-C data sets are undersampled and the interactions underestimated (discussed in for example Sati & Cavalli, Chromosoma 2017; Zhang *et al.*, Nat Comms. 2018; Montefiori *et al.*, eLife 2018; Cameron et al., 2019: <https://www.biorxiv.org/content/10.1101/377523>). Furthermore, interactions close to the viewpoint are overwhelmed by random interactions of sequences in close proximity and more difficult to detect. Nevertheless, the main points we intended to make are that cardiac-relevant contact maps are scarce, and that interaction/close proximity between promoters and putative REs does not imply the interaction is functional (it is only a requirement for function). We have revised the text of the Introduction accordingly.

9- "TADs are stable between different cell types and highly conserved between species"

Most TADs can be considered as stable among cells and species but this is not a universal rule. Around 30% of TADs show differences between cell types (Dixon et al., Nature 2012). Same for species.

To address this comment, we have rephrased the sentence: "TADs are mostly stable between different cell types and most boundaries are highly conserved between species (Dixon *et al.*, Nature 2012)". Moreover, we have used liver tissue of wildtype mice in the 4C-seq analysis of the *Scn5a* locus, using the *Scn5a* promoter and RE1 as viewpoint. The conformation of the locus was highly similar to that observed in heart (Supplementary Figure 7), indicating that at least for this locus, and these tissues, the TAD is conserved.

10- "These findings suggest that the enhancer cluster RE6-9 is partially redundant during development".

Partially redundant with which other elements? Partially referred to spatial or temporal pattern? Please state the interpretation more clear.

Please see also response to point 3. In the adult heart, deletion of the enhancer cluster RE6-9 causes loss of *Scn5a* expression from that allele. However, during development, homozygous embryos (E10.5) still express around 30% of *Scn5a* compared to wildtype controls. Furthermore, RE6-9 knockout mutants

survive up to E13.5, in contrast to homozygous *Scn5a* mutants, which die around E10.5. Please note that the entire enhancer cluster shows a developmental increase in H3K27ac in the heart (see revised Fig. 7), a histone modification associated with active enhancers, indicating the entire cluster becomes more active during development or after birth (see revised Fig. 7). This indicates that prior to the intermediate embryonic stage (E13.5) other regulatory elements (RE1, RE5) act during development to control *Scn5a* expression in the heart. We have rephrased the text for clarification.

11- “These findings suggest that RE6-8 is functionally redundant during development.”

Same as above. Clarification is also required here.

During development no change in *Scn5a* expression is observed in heterozygous and homozygous RE6-8 E13.5 embryos (revised Fig. 7e, g). In addition, the homozygous RE6-8 embryos appear normal and did not have any cardiac defects (revised Fig. 7c). This suggests that upon deletion of RE6-8 other regulatory elements (RE9, RE1 and RE5) act during development to control *Scn5a* expression in the heart. We have rephrased the text for clarification.

12- “Genetic variants affecting the activity of the enhancer cluster”

Conclusions and interpretation are missing in this section.

We found that rs6810361 is located at the edge of a Tbx20 binding site in RE9, which is occupied by Tbx20 in embryonic mouse hearts (revised Supplementary Figure 10). Our transfection assays in H10 (heart derived) cells indicates that the minor allele of rs6810361 reduces the activation of RE9 by Gata4 and Nkx2-5, two cardiogenic transcription factors that also interact with RE9. On the other hand, the response of RE9 to Tbx20 is reduced when rs6810361 is introduced. Recently, it has been shown that the minor allele (C) of rs6810361 decreases activity of a 325 bp fragment that is overlapping with the core of RE9 (including the Tbx20 binding site we have indicated), in mouse HL-1 cardiomyocytes under baseline conditions (Kapoor *et al.*, Proc Natl Acad Sci U S A. 2019). Together, these findings indicate that this single nucleotide variation might affect binding of other cardiac transcription factors, which results in a change in enhancer activity *in vitro*. We have included our interpretation and conclusion in section “Genetic variants affecting the activity of the enhancer cluster”.

13- “REs such as enhancers are usually found to act in a modular and additive fashion in that the each RE of a multi-component RE system adds its spatio-temporal activity pattern to that of the target gene(s)”

Many studies have shown that enhancers also act in different modes than just modular and additive. Authors should acknowledge these mechanisms and provide adequate references. This section of the discussion needs to be improved to frame the results in a better context with the current knowledge on the field.

We apologize for this oversight, indeed different modes of enhancer function are described in literature (Long *et al.*, Cell. 2016). To address this issue, we have mentioned the different modes of enhancer function in the Discussion section and provide adequate references. Furthermore, we have elaborated on

the potential function of the enhancer cluster RE6-9 in the transcriptional regulation of *Scn5a*, and its potential architecture and hierarchy.

14- “Recently, it was shown that such REs in the mouse genome act redundantly *in vivo*; individual removal of an RE from the genome causes only slightly reduced expression of the target gene that does not result in phenotypic changes”

Again, there are various example that show different modes of enhancer activity. See Gonen et al., *Science* 2018 or Shin et al., *Nature Genetics* 2016 (this last one is already cited by the authors). They are both examples of hierarchy within an enhancer cluster, like the case here presented.

We have addressed this issue in comment 13. In addition, we have provided additional examples of modes of enhancer activity (*e.g.* Gonen *et al.*, *Science* 2018, Hay *et al.*, *Nat. Genet.* 2016, Huang *et al.*, *Dev. Cell* 2016).

Minor comments:

1- Figure 2c would benefit from including *in situ* hybridizations for *Scn5a* and *Scn10a* to illustrate the similarities with the results of the enhancer reporter assays. In addition, the total number of embryos displaying positive staining and analyzed for RE6-9 should be indicated.

To address this comment, we would like to refer to van den Boogaard *et al.*, *JCI* 2012. In this paper, *in situ* hybridizations are included for *Scn5a* and *Scn10a* in E14.5 wildtype hearts, showing that the endogenous expression pattern resembles that of the *Scn5a* enhancer cluster RE6-9 analysed in the Hsp68/LacZ enhancer reporter assay (revised Fig. 2c). Furthermore, we have added the total number of embryos displaying positive staining and analyzed.

2- Please indicate in which specific reference can be located each of the transgenic reported displayed in Figure 1e. According to the cited references reporters were also performed for mouse RE9. It would be nice to include pictures side by side the human version of the tested enhancer, to appreciate the difference of activity between species, as well as to directly compare with the activity of RE6-9. Were also mouse enhancers tested for RE6 and RE7? If yes, please include it in the figure.

To address the first comment, Hsp68/LacZ enhancer reporter assay of human RE6 is shown in van der Harst *et al.*, *J Am Coll Cardiol.* 2016 and human RE9 is shown in van den Boogaard *et al.*, *JCI* 2014. These references have been added to the figure legend. Secondly, we have included a picture for mouse RE9 next to mouse RE6-9 and human RE9 in Reviewer Figure 3. Finally, we did not test the mouse enhancers RE6 and RE7 in Hsp68/LacZ enhancer reporter assays. We tested the 4 putative mouse REs (RE6-9) coupled together in one fragment (revised Fig. 2c).

Reviewer Figure 3. Dorsal view of E10.5 and E11.5 hearts containing human RE9, mouse RE9 and mouse RE6-9 in a *Hsp68/LacZ* enhancer reporter vector. Both human RE9 and mouse RE9 are strongly active in the interventricular septum region, and to a lesser extent in the atria and ventricles. A similar pattern was observed for mouse RE6-9. These sites of activity correspond to the sites of highest expression of *Scn5a* (and *Scn10a*) in the developing heart. Black scale bar is 0.5 mm.

3- “We found that the *Scn5a* promoter contacts multiple sites within the TAD including the promoters of *Scn10a* and *Exog*, RE6-9, RE5 within *Scn5a*, and RE1 within *Scn10a*”

It would be good for the readers to give a brief introduction of what is known about RE5 and RE1.

In the Introduction section, we have included information regarding RE1 located in an intron of *SCN10A* and RE5 located in an intron of *SCN5A* and included references describing this enhancer.

4- Figure 5a is referenced out of place in the text.

We have corrected the place of revised Fig. 5a in the text.

5- “Expression” is misspelled in Figs 7f and g

We have corrected the typo in revised Fig. 7f and g.

Reviewer #4

We thank the reviewer for insightful comments. We have addressed the points of the reviewer as detailed below.

I have some points for the authors' consideration.

1. Figure 1 could be more clearly presented. Multiple panels with some redundancy at present. The resolution is rather low for some panels also. Please consider highlighting the chromatin interactions that are of particular interest. Does the triangle plot, as I assume, represent LD in the region, or something else? Not labelled

We have revised Fig. 1 to make a clear overview of the human *SCN5A-SCN10A* locus. Panel a shows the GM12878 Hi-C heatmap (Rao *et al.*, Cell 2014) of the *SCN5A-SCN10A* locus and the TAD boundaries are marked. We have included PR interval (Verweij *et al.*, Circ Cardiovasc Genet. 2014; van Setten *et al.*, Nat Commun. 2018; and QRS duration variants (van der Harst *et al.*, J Am Coll Cardiol. 2016), an EMERGE track made from human epigenetic datasets (van Duijvenboden *et al.*, Nucleic Acids Res. 2016) and cardiac CTCF ChIP-seq track (ENCODE Project Consortium. Nature. 2012; Davis *et al.*, Nucleic Acids Res. 2018). Panel b depicts a zoom in of the human RE6-9 with GWAS variants (-log p-value) in this region, Mediator 1 ChIP-seq from human pluripotent stem cell-derived cardiomyocytes (Ang *et al.*, Cell. 2016) and H3K27ac of human left ventricle (Lister *et al.*, Nature. 2009, GSM908951). Lastly, panel c shows an Hsp68/LacZ enhancer reporter assay of human RE6 (van der Harst *et al.*, J Am Coll Cardiol. 2016) , RE7 and RE9 (van den Boogaard *et al.*, JCI 2014) in E10.5 mouse hearts.

2. Please make clear where the data underpinning Figure 1 originate. What for example is the input to the EMERGE predictions? It seems that many interactions are crossing the TAD domain, are the boundaries correctly represented? The scales on Figure 1 are not clear. It's stated that PR-interval variants are more found in SCN10A region and QRS variants more distributed, but this isn't easy to see, the statement needs statistical support. Is there H3k27Ac data available for hESC-CM's that could be integrated?

In revised Fig. 1a, we have used a Hi-C map of the lymphoblastoid cell line GM12878 (Rao *et al.*, Cell 2014) for the *SCN5A-SCN10A* locus which was adapted from the visualization tool HiGlass (Kerpedjiev *et al.*, Genome Biology 2018). Furthermore, we have included Supplementary Table 1 of human (and mouse) epigenetic datasets from a variety of cardiomyocyte (CM) containing tissues (both publicly available and unpublished), integrated in EMERGE as predictors for putative regulatory elements in the human *SCN5A-SCN10A* and mouse *Scn5a-Scn10a* locus. We have adjusted the scale of PR interval (Verweij *et al.*, Circ Cardiovasc Genet. 2014; van Setten *et al.*, Nat Commun. 2018) and QRS duration (van der Harst *et al.*, J Am Coll Cardiol. 2016) variants according to -log p-value. In revised Fig. 1b, we have included an H3K27ac track of human left ventricle (Lister *et al.*, Nature. 2009, GSM908951).

3. Figure 1 is focused on human data but there is also a mouse heart enhancer reporter assay. For clarity that would fit better in figure 2 which is mouse data.

The mouse heart enhancer reporter assay in revised Fig. 1d shows enhancer activity of human regulatory elements 6, 7 and 9. Therefore, we have decided to keep this panel in Fig. 1 as part of the human data.

4. In figure 2, there is substantial difference between the three transgenic enhancer examples. The lowest is quite different from the upper two. How is this to be interpreted? Please explain if the construct will only express LacZ if all the enhancer regions are active, or only one? Given the findings when the region is dissected, this is important to know.

We agree with the reviewer that there is a substantial difference in activity between the three transgenic enhancer examples in revised Fig. 2c. In each embryo, several copies of the construct are integrated at a random location in the genome, which affects the activity level and pattern of the enhancer. Therefore, each embryo displays a unique pattern. Only features of the pattern that are seen in 2 or more embryos are ascribed to the activity of the enhancer (heart-specific, interventricular septum region strongest). Our previous work has shown that individual regulatory elements (RE6 and RE9) in the human enhancer cluster RE6-9 were able to drive LacZ expression predominantly in the interventricular septum and to a lesser extent in the ventricles and atria of transgenic embryos (van den Boogaard *et al.*, JCI 2014; van der Harst *et al.*, J Am Coll Cardiol. 2016). A similar results was observed for the whole region RE6-9 as shown in revised Fig. 2c, indicating that these enhancers may all add to the pattern, and do not interfere with each other, as expected. We have clarified our interpretations in the revised Results section and addressed implications in the Discussion section.

5. The RE6-9 deletion was constructed so that the 3' ends of EXOG and SCN5A were not disrupted. I did not see confirmation that this was successfully achieved and those genes were left intact.

To address this comment, we have made a schematic representation of the deleted regulatory regions in the *Scn5a-Scn10a* locus and included the location of the TALENs and CRISPR sgRNAs that were designed to generate the various RE deletion models (revised Fig. 5a). Furthermore, genotyping primers for the RE6-9 TALEN deletion are designed at 3'ends of *Exog* and *Scn5a* (see below details genotyping primers, which confirms that both genes are left intact).

Position genotyping primers RE6-9 TALEN deletion: chr9:119374591-119392628 (mm9)

- Forward primer 5' CATGTCAGGTGTTGTTGAGGT 3'
- Reverse primer 5' CCCAGCGCTGTCGGATCAAATA 3'

6. Figures 3 a and b are interpreted to show that the RE6-9 hemizygotes exhibit a lesser degree of chromatin interaction from two viewpoints. The differences appear quite small, and there is no statistical test performed to support that the interactions are indeed different at the particular regions of interest between wildtype and hemizygote. I presume the greenish colour in the diagram indicates overlap but this is not stated.

We have revised Fig. 3a and b to show a better representation of the degree of chromatin interaction from two viewpoints. The new overlays are based on the signal at the level of single restriction fragment. The gray signal represents for the same contact profile for both the samples in the comparison. Furthermore, we have applied an unpaired student *t*-test to a number of locations using R 3.5.3. These results can be found in Supplementary Figure 3 for the *Scn5a* promoter viewpoint and Supplementary Figure 4 for the RE1 viewpoint. The analysis shows that intra-domain contacts between *Scn5a* promoter, *Scn10a* promoter, RE6-9, RE1 and RE5 are significantly reduced.

7. The allele specific expression experiment is presented in figure 5b. This must, I think, represent sequence of expressed RNA, but it's not stated. It's something of a concern that the wildtype FVB alleles at *Scn5a* appear to have lower expression than the wildtype SPRET alleles, in the top panel, though it is convincing that they are undetectable in the FVB KO, which isn't the case for the other genes. Please clarify and address.

We have clarified our approach in the allele specific expression experiment in the Results and Methods sections. To assess the allele specific expression of *Scn5a* we compared the genome of FVB/NRj strain with the genome of other mouse strains for sequence variations using www.sanger.ac.uk/science/data/mouse-genomes-project. We concluded that of the available strains, only SPRET contained sufficient sequence variation in *Scn5a*, *Scn10a* and *Exog*. We used RNA isolated from the left ventricle as material for Sanger sequencing.

The difference in ratio of peak heights between wildtype FVB allele and wildtype SPRET allele at *Scn5a* might be caused by technical issues during sequencing. For example, variations elsewhere in the amplicon can affect PCR amplification efficiency that results in molar variant ratios (Carr *et al.*, Bioinformatics. 2009). Furthermore, context-dependent dye-terminator dideoxynucleotide incorporation can result in variation of peak heights (Carr *et al.*, Bioinformatics. 2009).

8. How do the authors interpret the lack of effect of RE6-9 deletion in the right atrium (Figure 5c)?

We were surprised to observe a lack of effect of RE6-9 in the right atrium. We speculate that the RE6-9 enhancer is not required to drive expression in the right atrial compartment. It could be that other enhancers such as RE1, or the promoter of *Scn5a* act dominantly in the right atrium. We have seen a similar phenomenon when deleting an enhancer absolutely required for *Nppa-Nppb* expression in the ventricles. Expression of these genes in the atria is hardly affected, as their expression in this compartment seems to be driven by the promoters of these genes. We have provided an interpretation in the revised Discussion.

9. In the *in vivo* experiments, the difference between the PR intervals in the wildtype and hemizygous animals in the pseudo-ECG Langendorff experiment appears to be about 30%, but *in vivo* it is only 3%, a perhaps surprisingly small difference. Please place these differences in context.

In the *ex vivo* situation autonomic modulation is absent, which is the major difference with the *in vivo* situation. Although AV conduction is not slower *ex vivo* than *in vivo* (Cardiovasc Res. 2013 Jan 1;97(1):182-91) we think the balance between parasympathetic and sympathetic tone is an important determinant of

AV conduction. This latter is exemplified by the PR shortening effect of atropine (muscarine receptor blocker) and PR prolonging effect of metoprolol (betablocker) in mice *in vivo* (Merentie *et al.*, *Physiol Rep.* 2015). We speculate the autonomic balance maintains safety of AV conduction in affected AV nodes, like in our RE6-9 mice. Decreased adrenergic tone - as is the case in *ex vivo* preparations - can be achieved using betablockers. In mice harboring the deltaKPQ-Scn5a mutation, betablockers have a larger effect on AV conduction than in control mice (Fabritz *et al.*, *Cardiovasc Res.* 2010, Pages 60–72). The latter supports our hypothesis that autonomic modulation is of crucial importance for AV conduction in the setting of reduced sodium current or sodium channel dysfunction. We have added the following to the Result section: *We speculate the autonomic balance maintains safety of AV conduction in affected AV nodes of RE6-9^{+/-} mice in vivo, providing an explanation for the differences in PR prolongation in in vivo and ex vivo ECG recordings* (page 8, line 249-251).

10. Rs6810361 is a QRS duration associated variant, is it also a Brugada Syndrome associated variant? Why did the authors test a QRS duration variant when their mouse model had shown no effect on QRS duration but rather PR interval?

Common variant rs6810361 is associated with PR interval based on the latest PR interval genome-wide association meta-analysis (van Setten *et al.*, *Nat Commun.* 2018). Furthermore, we have performed additional ECG recordings in RE6-9^{+/-} mutants, which revealed lengthening of both PR interval and QRS duration when compared to wildtype mice (revised Fig. 6).

Reviewers' comments:

Reviewer #1 (Remarks to the Author):

Although I could quibble further on some details of the some of the studies, overall I am satisfied by the authors responses and have little to add.

Reviewer #3 (Remarks to the Author):

I would like to thank the authors for the extensive work performed, which substantially improved the quality and clarity of the manuscript. In particular, evaluating the effects of RE6-9 deletion in the context of the transcriptional adaptation provides valuable clues on interpreting enhancer function. Overall, this present study represents an important piece of work that will contribute substantially to a better understanding of 3D-gene regulatory mechanisms. Nevertheless, I have some additional comments that I summarize below:

- The authors made an exhaustive reanalysis and show very convincingly that the deletion of RE6-9 disrupts the 3D-organization of the Scn5a locus. This effect is evident from the observed interactions between the Scn5a promoter and the Scn10 promoter, RE5 and RE1. The authors also measure decreased interactions with RE6-9 itself but a technical issue questions the validity of this result.

The authors define a genomic location at RE6-9 (position 119,372,000) and extend it 5 Kb up and downstream. They use the fragments included in this window to compare the reads obtained in 4C-seq experiments between mutants and controls. This approach would include fragments in the genomic region that ranges from 119,367,000 to 119,377,000

However, the heterozygous deletion of RE6-9 (TAL3+4, TAL5+6) ranges from 119,374,891 to 119,392,359, which overlaps partially with the interval delimited for the 4C-seq differential analysis. This would result in the inclusion of fragments that, in mutants, would render no interactions from one of the alleles. Therefore, the number of total interactions measured in the mutants would be underestimated.

To solve this problem, the data could be reanalyzed using genomic coordinates that are slightly shifted upstream so that these fragments are effectively not included.

-Graphs in Supplementary Figures 3 and 4 are truncated. Please correct.

- The genomic positions of TAL3+4 and TAL5+6 are mixed up in Supplementary Table 4 related to Figure 5.

- A few items are cited out of order across the manuscript.

Reviewer #4 (Remarks to the Author):

The authors have made satisfactory responses to the points raised in my original review. As per my previous assessment, this is an interesting study with important findings.

Response to Reviewers' comments: NCOMMS-18-35532A

Reviewer #1 (Remarks to the Author):

Although I could quibble further on some details of the some of the studies, overall I am satisfied by the authors responses and have little to add.

We thank the reviewer for the positive evaluation of the revised manuscript.

Reviewer #3 (Remarks to the Author):

I would like to thank the authors for the extensive work performed, which substantially improved the quality and clarity of the manuscript. In particular, evaluating the effects of RE6-9 deletion in the context of the transcriptional adaptation provides valuable clues on interpreting enhancer function. Overall, this present study represents an important piece of work that will contribute substantially to a better understanding of 3D-gene regulatory mechanisms.

Nevertheless, I have some additional comments that I summarize below:

We thank the reviewer for the encouraging evaluation, careful reading and constructive comments. Please find below a detailed response to all points.

Comments

The authors made an exhaustive reanalysis and show very convincingly that the deletion of RE6-9 disrupts the 3D-organization of the Scn5a locus. This effect is evident from the observed interactions between the Scn5a promoter and the Scn10 promoter, RE5 and RE1. The authors also measure decreased interactions with RE6-9 itself but a technical issue questions the validity of this result. The authors define a genomic location at RE6-9 (position 119,372,000) and extend it 5 Kb up and downstream. They use the fragments included in this window to compare the reads obtained in 4C-seq experiments between mutants and controls. This approach would include fragments in the genomic region that ranges from 119,367,000 to 119,377,000. However, the heterozygous deletion of RE6-9 (TAL3+4, TAL5+6) ranges from 119,374,891 to 119,392,359, which overlaps partially with the interval delimited for the 4C-seq differential analysis. This would result in the inclusion of fragments that, in mutants, would render no interactions from one of the alleles. Therefore, the number of total interactions measured in the mutants would be underestimated. To solve this problem, the data could be reanalyzed using genomic coordinates that are slightly shifted upstream so that these fragments are effectively not included.

To address this issue, we have reanalyzed the interactions after slightly shifting the window upstream of position 119,372,000, as suggested by the reviewer. This 10 Kb window (from position 119,363,500 to 119,373,500) does not overlap with the deletion. This reanalysis shows that the interaction between RE6-

9 and the *Scn5a* promoter (Supplementary Figure 5), and between RE6-9 and RE1 (Supplementary Figure 7) is significantly different between the RE6-9^{+/-} mice and wildtype controls.

- Graphs in Supplementary Figures 3 and 4 are truncated. Please correct.

We have adjusted the y-axis range of the 4C plots in Supplementary Figures 5 and 7 (previously Supplementary Figure 3 and 4).

- The genomic positions of TAL3+4 and TAL5+6 are mixed up in Supplementary Table 4 related to Figure 5.

We have corrected the genomic positions of TAL3+4 and TAL5+6 in revised Supplementary Table 4 according to Figure 5a.

A few items are cited out of order across the manuscript.

To address this issue, we have thoroughly checked the manuscript and cited the items in order related to the text.

Reviewer #4 (Remarks to the Author):

The authors have made satisfactory responses to the points raised in my original review. As per my previous assessment, this is an interesting study with important findings.

We thank the reviewer for the positive assessment of the revision.

REVIEWERS' COMMENTS:

Reviewer #3 (Remarks to the Author):

The authors have successfully addressed all my remaining concerns.